# Developmental changes in story-evoked responses in the neocortex and hippocampus

**Samantha S Cohen\*, Nim Tottenham, Christopher Baldassano**

Department of Psychology, Columbia University, New York, United States

**Abstract** How does the representation of naturalistic life events change with age? Here, we analyzed fMRI data from 414 children and adolescents (5–19 years) as they watched a narrative movie. In addition to changes in the degree of inter-subject correlation (ISC) with age in sensory and medial parietal regions, we used a novel measure (between-group ISC) to reveal age-related shifts in the responses across the majority of the neocortex. Over the course of development, brain responses became more discretized into stable and coherent events and shifted earlier in time to anticipate upcoming perceived event transitions, measured behaviorally in an age-matched sample. However, hippocampal responses to event boundaries actually decreased with age, suggesting a shifting division of labor between episodic encoding processes and schematic event representations between the ages of 5 and 19.

## Editor's evaluation

Cohen et al., present analyses of a large publicly available set of neuroimaging data from children and adolescents watching an animated video, and is likely to be of interest to neuroscientists interested in methods for analyzing naturalistic neuroimaging data, or those interested in the development of narrative processing in the brain. The methodological approach developed here is a valuable addition to the repertoire of developmental neuroscience.

**\*For correspondence:**
samantha.s.cohen@gmail.com

**Competing interest:** The authors declare that no competing interests exist.

## Introduction

The ability to perceive and remember the world changes radically throughout the first twenty years of life (*Aslin and Smith, 1988*; *Schneider and Pressley, 2013*). Older children and adolescents are better able to understand and interpret the world around them, and anticipate upcoming situations (*Carpendale and Lewis, 2006*; *Casillas and Frank, 2017*; *Richardson and Saxe, 2020*). Much of this increase in ability can be attributed to greater familiarity with the events that they are likely to encounter (*Schneider and Pressley, 2013*). For instance, children have better memory for situations that they have expertise in (e.g. Chess; *Chi, 1978*). Here, we characterize changes that occur in the brain's response to complex naturalistic stimuli during this period of the acquisition of structured knowledge about the world.

Previous research has examined developmental changes in neural activity when learning principled relationships between items in a laboratory setting (*Brod et al., 2017*). Children, however, normally acquire and organize novel information about their world over the course of weeks, if not years (*Nelson, 1986*). Naturalistic narrative stimuli provide a tool for probing this complex, real-world knowledge that children have acquired across repeated experiences and can deploy automatically (*Cantlon and Li, 2013*; *Lerner et al., 2019*; *Petroni et al., 2018*; *Moraczewski et al., 2018*; *Moraczewski et al., 2020*; *Richardson et al., 2019*; *Richardson and Saxe, 2020*; *Vanderwal et al., 2020*).

We utilized functional magnetic resonance imaging (fMRI) data acquired while children and adolescents between the ages of 5 and 19 watched a short narrative cartoon that contained both social and emotional content (*Alexander et al., 2017*; *Petroni et al., 2018*). Although cartoons do not physically replicate the characteristics of everyday perception, there is evidence to suggest that children respond similarly to the social cues in cartoons and live action videos (*Han et al., 2007*). Analyzing brain responses to complex stories or movies is challenging, since we do not yet have models that can predict brain-wide responses to these types of stimuli, especially given the fact that the representation of meaning likely changes with age (*Clark, 1973*). A model-free approach to assess brain responses to this kind of stimuli is inter-subject correlation (ISC), which measures the similarity of brain responses in a brain region across movie viewers (*Cantlon and Li, 2013*; *Lerner et al., 2019*; *Petroni et al., 2018*; *Moraczewski et al., 2018*; *Moraczewski et al., 2020*). ISC within an age group has generally been found to increase with age, possibly due to more mature engagement with the content, greater shared knowledge about the world, or more exposure to widely used cinematic conventions (*Cantlon and Li, 2013*; *Lerner et al., 2019*; *Moraczewski et al., 2018*; *Moraczewski et al., 2020*). We therefore measured ISC in parcels throughout the cortex (*Schaefer et al., 2018*) to assess if processing becomes more similar with age. In line with previous studies, we hypothesized that the magnitude of ISC would increase with age, although decreases have been found in studies in other modalities (*Petroni et al., 2018*).

ISC has also been measured between age groups to assess the similarity between the responses of children and adults (*Cantlon and Li, 2013*; *Lerner et al., 2019*; *Moraczewski et al., 2018*). If a child is more similar to adults, they are considered more mature, and therefore likely to have better academic or social abilities (*Cantlon and Li, 2013*; *Moraczewski et al., 2018*). This previous work is consistent with the idea that children are simply noisy versions of adults, and the noise level decreases with age. We test an alternative hypothesis: that children and adolescents have *different* average response timecourses, corresponding to different age-related interpretations of the movie. We therefore employed a new across-group ISC measure that allows for comparisons across age groups while controlling for the degree of similarity within each age group. This allowed us to determine whether there was a consistent shift in the group-level temporal patterns of activity, independent of changes in within-group consistency. We hypothesized that in default mode network regions responsible for story interpretation and self-referential thought, even where within-age ISC magnitude does not change, the pattern of activity representing the movie will change across development, just as the semantic interpretation of movies changes with age (*Nelson, 1986*; *Raichle et al., 2001*).

How can we characterize the changes that are occurring in response timecourses? Although some knowledge of the hierarchical structure of the events that compose a narrative develops in infancy, the ability to reliably notice these events does not mature until at least the teenage years (*Yates et al., 2021*; *Zacks and Tversky, 2001*; *Zheng et al., 2020*). It is likely that the characterization of events changes with age. We therefore asked both children and adults to subjectively report where they believed meaningful scene changes occurred in the narrative. We hypothesized that although there would be no change in the behaviorally reported location of these coarse-grained narrative segments, the neural representation of events along the cortical hierarchy would change with age.

Recent work in adults has shown that the neural activity evoked by narratives are characterized by stable patterns of responses, and that the moments of transition between stable patterns in association areas correspond to these meaningful boundaries between 'events'' in a continuous perceptual stream (*Baldassano et al., 2020*; *Baldassano et al., 2018*; *Chen et al., 2017*). These event patterns are driven in part by learned schematic scripts about common experiences in the world, such as the sequence of events that characterize a visit to a restaurant (*Baldassano et al., 2018*).

We hypothesized that with age, the strength of schematic event representations, that are able to generalize across different instances of similar events, will increase due to more experience with different exemplars. These kinds of representations should be stored in default mode regions such as the medial prefrontal cortex and posterior medial cortex (PMC). Following previous research in adults, we assessed the ability of children of different ages to segment their world into discrete chunks with a Hidden Markov model (HMM; *Baldassano et al., 2020*; *Baldassano et al., 2018*; *Lee et al., 2021*). We predicted that activity patterns will be more consistent among older subjects and become more stable across timepoints within events, providing a better fit to the HMM event segmentation model.

If we do not find a general improvement in event model fits with age, this will refute the hypothesis that the schematic event representations that support event models are strengthened with age.

It is also likely that the timing of event transitions changes as a function of age. Activity in regions responsible for representing the viewpoints of others occurs earlier during the second presentation of a movie in children between six and seven years then in children between three and four years (*Richardson and Saxe, 2020*). Here we test whether, on an *initial* viewing of a naturalistic video, the timing of event transitions varies with age. Unlike previous work that considered anticipation at only a fixed offset of two seconds (*Richardson and Saxe, 2020*), the HMM-derived transition timings provide a flexible approach for detecting timing shifts between age groups that can vary across brain regions with different processing timescales (*Baldassano et al., 2020*; *Hasson et al., 2015*) and can vary across timepoints throughout the movie. In line with the idea that schematic event representations improve with age, older adolescents should be able to anticipate events further into the future due to their increased experience with the world.

Another characteristic of adult responses to narrative movies is the robust hippocampal activity evoked by boundaries between events (*Baldassano et al., 2020*; *Ben-Yakov and Henson, 2018*). Age-related changes in this signal have been previously observed in older adults, with response decreases between ages 18 and 88 in the posterior hippocampus (*Reagh et al., 2020*). It is therefore possible that development of event-structured responses in the cortex is mirrored by changes in the hippocampus's ability to respond to event transitions. We predicted that the magnitude of the hippocampal response to event boundaries would increase with age, in line with a model of maturation wherein the ability to encode the unique episodes of daily experience increase into middle age and then decreases with senescence (*Grady, 2012*; *Reagh et al., 2020*). However, should we find a decrease in the hippocampal response to event boundaries with age, this would provide support

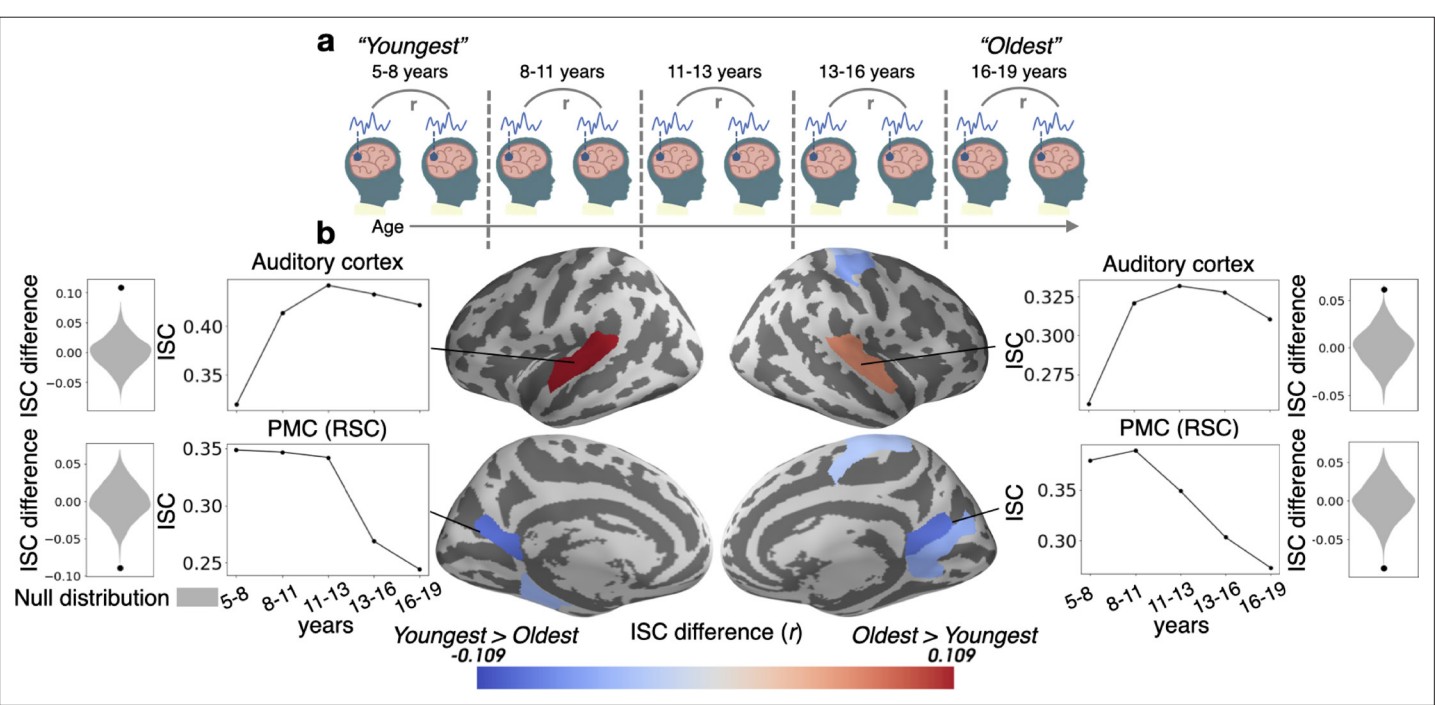

**Figure 1.** Inter-subject correlation (ISC) increases with age in sensory regions and decreases with age in the PMC. (**a**) Within-group ISC was computed for five distinct age groups, and a statistical comparison was performed between the Oldest and Youngest age groups. (**b**) The difference in ISC between the Youngest (5–8years) group and the Oldest (16–19years) group is displayed in significant parcels (q<0.05) on the cortical surface. ISCs for all age groups are plotted for four significant parcels, selected post hoc for illustration, along with the ISC difference between the Youngest and Oldest groups compared to the null distribution.

The online version of this article includes the following figure supplement(s) for figure 1:

**Figure supplement 1.** Inter-subject correlation (ISC) across a random mixture of the Youngest and Oldest subjects.

**Figure supplement 2.** Inter-subject correlation (ISC) within the Youngest and Oldest subjects.

**Figure supplement 3.** The demographic breakdown of the subject groups studied.

for the idea that younger children may focus on the episodic encoding processes responsible for encoding the specific details of events as they work towards creating more stable schematic event representations (*Keresztes et al., 2018*; *Maril et al., 2010*).

## Results

We sought to uncover whether and how the neural representation of naturalistic stimuli change with age between 5 and 19 years. To accomplish this, we used a large, publicly available dataset (*Alexander et al., 2017*) of functional magnetic resonance imaging (fMRI) data recorded while children watch a short video animation with both social and emotional themes. As the data were not equally distributed with age, the results are derived from equally sized subsamples from each age group (each with an age span of approximately 3 years). To maximize the power and reproducibility of the results, our analyses were averaged across five random subsamples from each age group (See Methods; *Poldrack et al., 2017*).

We first measured inter-subject correlation (ISC) across all subjects, as well as within the youngest (5–8 years) and oldest (16–19 years) age groups, as a general measure of story comprehension and engagement (*Figure 1—figure supplement 1* and *Figure 1—figure supplement 2*; *Nastase et al., 2019*). To statistically examine changes in this measure due to age, we compared ISCs for the youngest and oldest age groups, calculated within parcels (*Schaefer et al., 2018*). We found (*Figure 1*) that ISC increases with age in low level sensory regions such as the auditory cortex, and decreases with age in some higher level association regions such as the retrosplenial cortex (RSC) portion of the PMC. To ensure that the increases in ISC with age are not due to differences in the level of noise between the groups, we measured the relationship between the framewise displacement of each child in the Youngest group and their ISC with the other subjects. There was no relationship between framewise displacement and ISC in any of the parcels where ISC increased with age, indicating that motion did not drive the result in these parcels (all q's>0.05).

In most of the cortex, there is no difference in ISC magnitude as a function of age, which could indicate that children of all ages engage similarly with the movie (*Petroni et al., 2018*). It is also possible, however, that responses are highly consistent within each age group but differ systematically across age groups. For instance, it is possible that the most salient features of the video change with

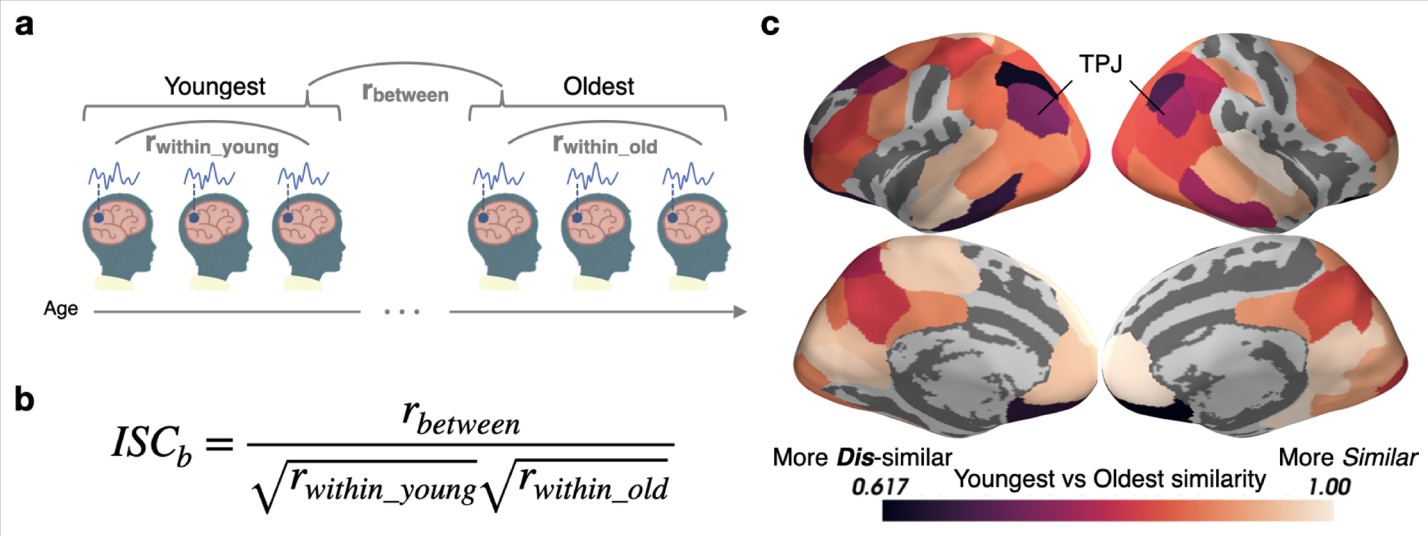

**Figure 2.** Stimulus processing differs in semantic and default mode regions between the Youngest and Oldest ages. (**a**) Response timecourses were correlated within the Oldest and Youngest groups separately, and also correlated between groups. (**b**) For each parcel, between-group ISC ($ISC_b$) is calculated by dividing the across-group correlation by the geometric mean of the within-group correlation. This allows us to identify regions in which the correlations between groups are smaller than we would expect based on the within-group similarities, reflecting differing mean responses across groups. (**c**) The Youngest and Oldest groups are most dissimilar (indicated by darker colors) in regions in the default mode network, such as the temporoparietal junction, and in regions responsible for semantic processing, such as the inferior temporal cortices, temporoparietal junction, and dorsolateral prefrontal cortex (parcels shown have $ISC_b$ values less than age-shuffled null permutations, thresholded at q<0.05).

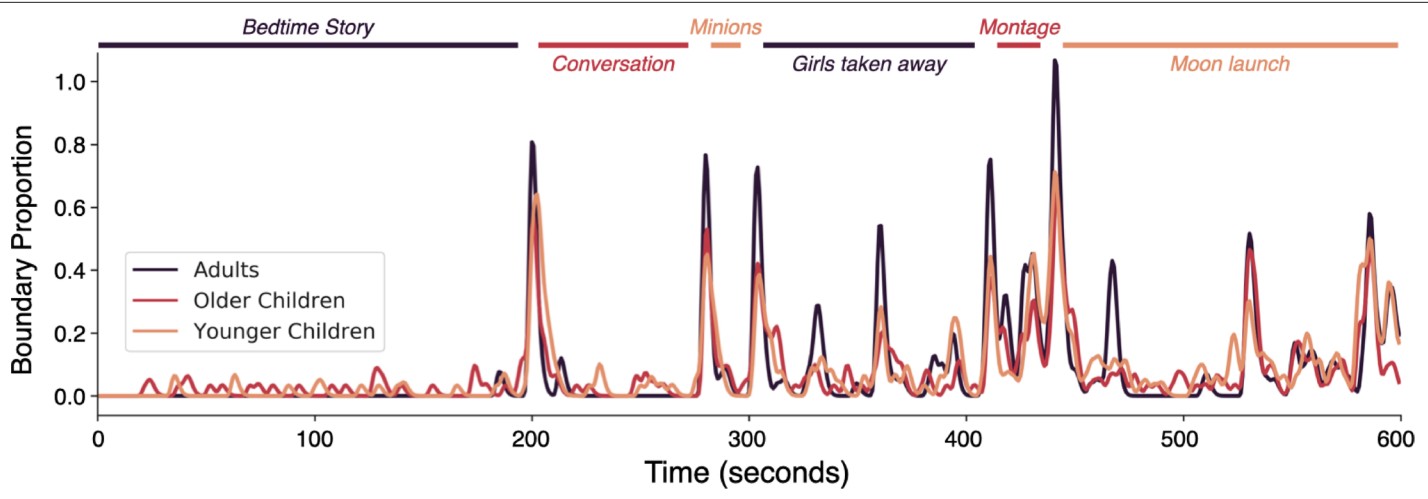

**Figure 3.** Event boundary alignment between children and adults. The proportion of Adult, Older Children, and Younger Children raters who marked a boundary during the stimulus. Colored lines indicate the event durations and boundaries agreed upon by at least half of a group of independent adult raters (see Methods). Brief descriptions of each scene are written in italics.

age, eliciting responses that are predictable within an age group but different across groups. To get a pure measure of across-group similarity that is insensitive to within-group variation, we use a novel measure called between-group ISC (ISC$_b$; see Appendix 1) (**Figure 2a**). To calculate the correlation between groups, ISC$_b$ is computed by dividing the between-group (Youngest to Oldest) correlation by the geometric mean of the within-group correlation (**Figure 2b**).

Eighty-one of the 100 (**Schaefer et al., 2018**) parcels tested show significant age-specific response timecourses including default mode regions, such as the temporoparietal junction (TPJ) and PMC (**Raichle et al., 2001**), and language and concept sensitive regions, such as the dorsolateral prefrontal cortex (dlPFC) and inferior temporal cortices, and orbitofrontal cortex (**Binder and Desai, 2011**; **Ralph et al., 2017**; **Figure 2c**).

This result demonstrates that responses are changing substantially with age in many brain regions, but does not indicate *how* these responses are changing. One possibility is that the interpretation of the events and scenes in the narrative changes with age. We therefore assessed when adults and children, age-matched to the fMRI sample, report that an event in the narrative has finished, and a new event has begun. There are small but significant differences in the timing of the event boundaries marked by adults and children (ISC$_b$ = 0.97, p=0.009). This discrepancy was true for both the younger and older children, separated by a median split based on age (adult-older children ISC$_b$ = 0.95, adult-younger children ISC$_b$ = 0.91, both p's<1 × 10$^{-5}$). The timing of event boundaries was also slightly, but significantly different between the younger half and older half of the children (ISC$_b$ = 0.96, p=0.01, **Figure 3**). Given this overall similarity in behavior, we next ask whether there are differences in how these events are represented and tracked in the brain.

In adults, movie responses are often characterized by rapid transitions between stable periods of brain activity, corresponding to meaningful events in the narrative (**Baldassano et al., 2020**; **Chen et al., 2017**). To identify the extent of this temporal clustering in our data, and the timescale of this clustering, we use a Hidden Markov Model (HMM) (**Baldassano et al., 2020**). Fitting the HMM separately to the Youngest and Oldest groups, we found that the HMM was able to identify event structure in at least one of the groups (Youngest and/or Oldest) for 70 out of our 100 parcels (**Figure 5—figure supplement 1**). In these 71 parcels, we found that the optimal timescale (optimal number of events) was highly correlated between the Youngest and Oldest ages (r=0.78, RMS difference between groups = 12.3 s), showing increasing timescales from sensory regions to higher level association regions (**Figure 5—figure supplement 2**). This replicates previous research showing increasing event timescales across the cortical hierarchy (**Baldassano et al., 2020**; **Geerligs et al., 2021**; **Hasson et al., 2015**), and shows that this timescale hierarchy is present even in our Youngest (5–8 years) age group.

As all 71 parcels had similar timescales in both groups, we jointly fit an HMM (with the parcel's optimal number of events) simultaneously to the Youngest and Oldest groups. As expected based

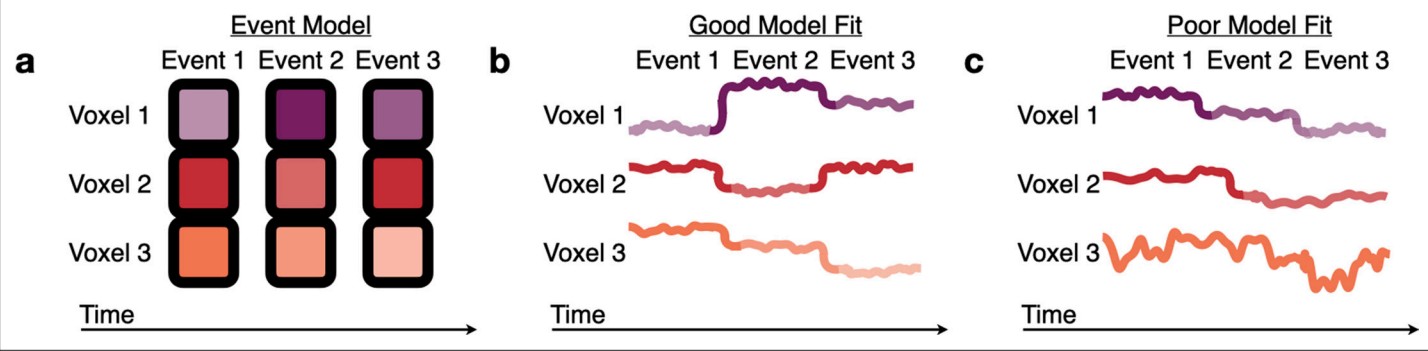

**Figure 4.** Measuring fits to the HMM event model. (**a**) The Hidden Markov Model (HMM) assumes that brain activity in response to a movie should proceed through a specific sequence of stable event patterns, each with a specific pattern of high and low activities across voxels (represented here as the saturation of each color). (**b**) The model is a good fit to brain responses that exhibit patterns consistent with the model assumptions, sequentially transitioning between the HMM event patterns with little variability during events. (**c**) A poor model fit indicates that this event model does not capture a brain region's dynamics, because the order of the relative activity levels does not match the model's sequence of event patterns (Voxel 1), the event transitions do not align between voxels (Voxel 2), or there is high within-event variability (across time or across subjects) (Voxel 3).

on previous research in adults, the event boundaries from this model correspond to the boundaries subjectively placed by human raters in association regions, such as PMC, TPJ, and precuneus (*Figure 5—figure supplement 3*; *Baldassano et al., 2020*). The jointly fit HMM produced an ordered set of event patterns which were shared across both groups (*Figure 4a*). We then tested this event model in held-out subjects, allowing us to measure in each group: (1) the model fit (log likelihood on held-out data), indicating the extent to which brain responses could be explained as an ordered sequence of these stable event patterns (*Figure 4b–c*), and (2) the timing of event transitions. We hypothesized that the Oldest subjects would have a better model fit than the Youngest subjects, indicating increased stability and reliability of event representations. We also hypothesized that the

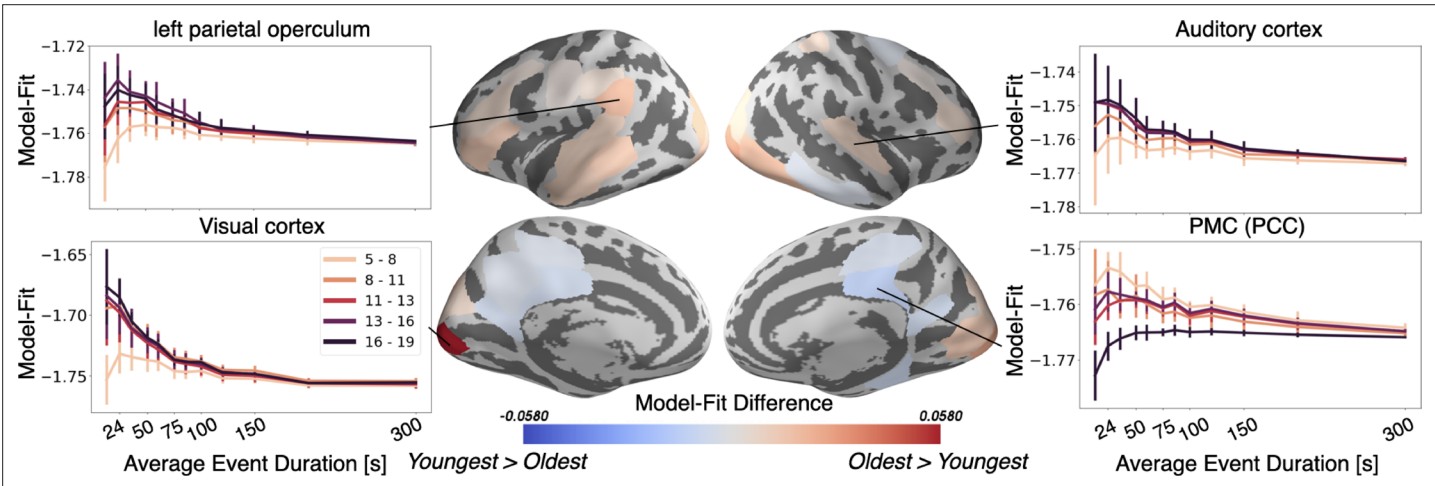

**Figure 5.** The Oldest group has better event models in regions including visual cortex and auditory cortex, whereas the Youngest group has a better model-fit in PMC. Regions with a significant model-fit difference between the youngest and oldest age group (at the optimal timescale for that region) are plotted on the cortical surface. Redder shades indicate that the event model fits improve with age, and bluer shades indicate that they weaken with age. The model fits for event models with different event durations are shown across all age groups and event durations in four example regions, selected post hoc for illustration. Error bars represent the standard deviation of model fit in the held out subjects, averaged across five cross-validated folds.

The online version of this article includes the following figure supplement(s) for figure 5:

**Figure supplement 1.** Comparing the model-fit difference for all parcels between the Youngest and Oldest ages.

**Figure supplement 2.** The best-fitting average duration of events for the optimal HMMs trained and tested on either the Youngest or the Oldest ages.

**Figure supplement 3.** The HMM-derived event boundaries correlate with behaviorally estimated event boundaries from children.

Oldest group would exhibit earlier event transitions than the Youngest group, due to predictive, anticipatory, or preparatory processes (*Carpendale and Lewis, 2006*; *Lee et al., 2021*; *Richardson and Saxe, 2020*).

As hypothesized, the neural activity of the Oldest group can be better represented by an event model in most of the parcels that significantly differ with age. These regions include sensory areas, such as the auditory and visual cortex, as well as the left parietal operculum and dlPFC (*Figure 5*). In sensory regions, the largest change in event model-fit occurs between children 5–8 years and those 8–11 years (*Figure 5*, side panels). The left parietal operculum increases in model-fit through approximately age 13. These developmental trajectories hold not just for these regions' optimal event timescale, but also across a wide range of event timescales (*Figure 5*, side panels). Surprisingly, the HMM exhibits better model fits in the PMC, including both the posterior cingulate cortex (PCC) and RSC, for the Youngest group than the Oldest group. These results partially reflect changes in across-subject

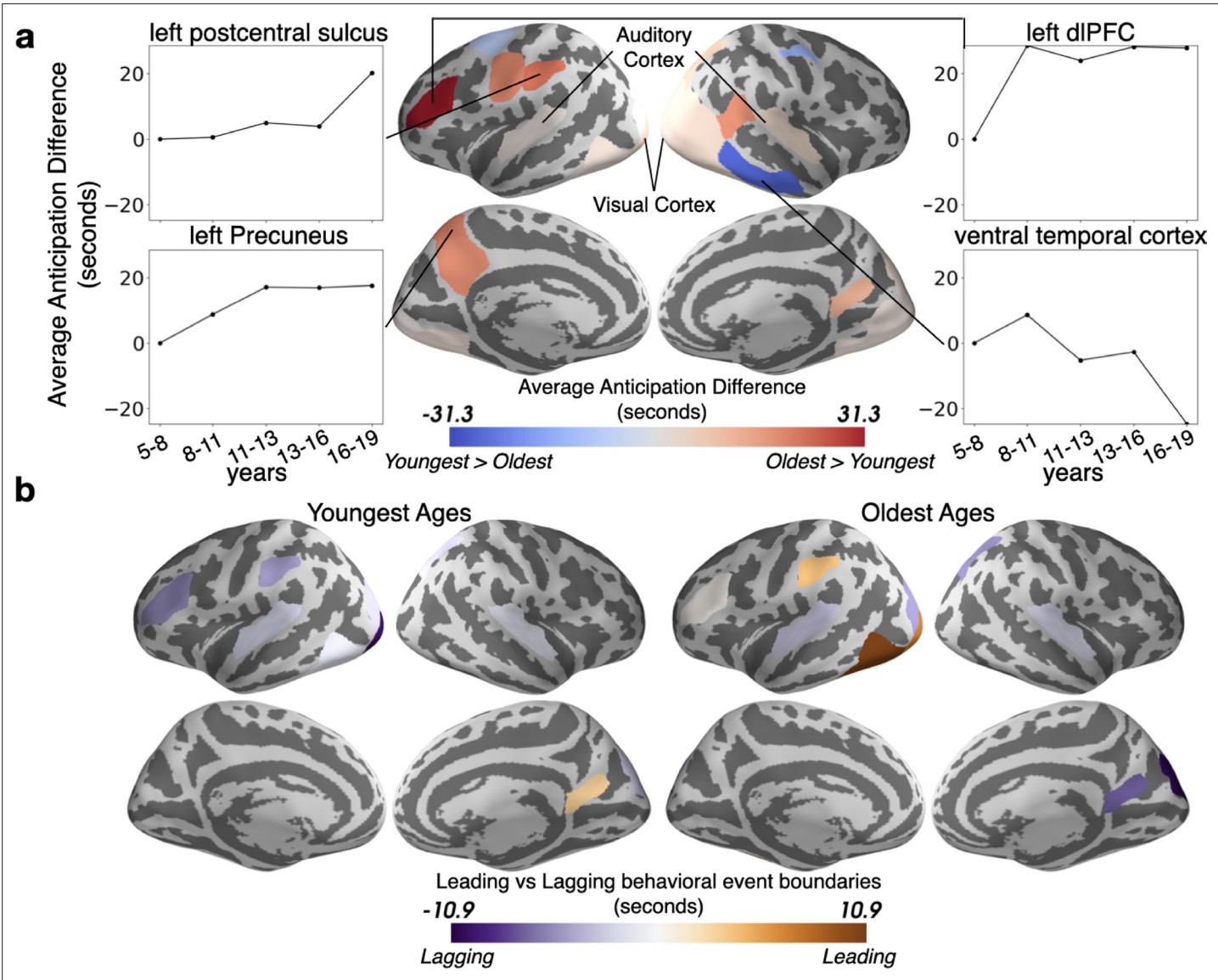

**Figure 6.** The timing of event boundaries shifts with age. (**a**) The Oldest group represents upcoming events before the Youngest group in auditory and visual cortex, as well as left lateralized associative regions including the left precuneus and left dorsolateral prefrontal cortex (reddish hues). The Youngest group anticipates events sooner in the right ventral temporal cortex (bluish hues). Highlighted parcels showing age anticipation trajectories were selected post hoc for illustration. (**b**) The event boundaries in the brains of the Oldest group generally lead behaviorally-derived event boundaries, whereas the transitions in the brains of the Youngest group generally lag behind behavior. In both (**a**) and (**b**), the parcels shown have differences greater than age shuffled permutations (q<0.05).

consistency as identified by the model-free ISC analysis (*Figure 1*), but also show effects in additional regions including the dlPFC.

We also expected that event boundaries would occur earlier in time with age, because adolescents are more experienced with real-world schemas and can thus predictively represent upcoming situations. This hypothesis was borne out in sensory regions including auditory and visual cortex, as well as left lateralized associative regions including the left precuneus and left dlPFC (*Figure 6a*). Sensory regions are predictive at shorter time scales, consistent with their quicker processing speed (*Baldassano et al., 2020*; *Hasson et al., 2015*; *Lee et al., 2021*), while the higher level association regions anticipate upcoming events on average up to tens of seconds into the future. This anticipation increases rapidly across ages 5–11 in the left dlPFC and left precuneus, with a more variable trajectory in left postcentral sulcus (side panels). However, as was seen for model-fit, this trend reverses in the right ventral temporal cortex where the Youngest children anticipate upcoming events slightly sooner, and anticipation decreases almost linearly with age.

To determine whether developmental changes in the timing of event transitions reflect an increased level of anticipation in the Oldest group, or, alternatively, a delayed response in the Youngest group, we compared the HMM-derived event boundaries to event transitions behaviorally identified by an age-matched group of children. Across the cortex we find that the Oldest ages generally anticipate event shifts, while the Youngest ages lag behind them to some degree (*Figure 6b*).

Finally, we asked whether hippocampal responses differ as a function of age at the event boundaries reported by our age-matched sample of children. We found a robust response to event boundaries in all ages (all p's<0.0001 for each age group's response at time 0, see *Figure 7a*, right side). This response decreases with age between 5 and 19 years old ($r=0.{-}16$, $p=1 \times 10^{-3}$, N=414; *Figure 7a*). As previous research has indicated that the anterior and posterior HPC may potentially respond differently to event boundaries (*Reagh et al., 2020*), we examined whether the anterior or posterior HPC were driving this change in boundary-driven response with age. We found that there is a significant decrease in the anterior hippocampal (aHPC) response to event boundaries with age ($r={-}0.18$, $p=2 \times 10^{-4}$, *Figure 7b*). We did not find a significant decrease in the posterior HPC (pHPC; $r={-}0.09$, $p=0.06$, *Figure 7c*), nor a significant interaction between HPC subregions ($p=0.7$). In none of these regions was the change in boundary-related response explained by anatomical volume changes (see Methods).

## Discussion

Using a novel measure of response similarity, we found that the processes used to perceive a dynamic narrative change from ages 5–19 throughout the majority of the neocortex and in the hippocampus. These developmental changes are characterized by increasingly event-structured dynamics in the neocortex, with periods of stability punctuated by rapid transitions, and anticipatory shifts in event timing. In parallel, hippocampal responses at event boundaries *decrease* in magnitude with age. These changes suggest that the brain's strategy for narrative processing shifts from encoding each event as a novel episode in the hippocampus to activating and maintaining generalized event knowledge in the cortex, storing only episode-specific information into long-term memory.

Older ages exhibit more across-subject synchrony in auditory cortex (*Figure 1*), consistent with prior work showing developmental increases in ISC evoked by other similarly highly produced animated movies (*Moraczewski et al., 2018*). These results are likely related to more strongly correlated semantic processing in adolescents, resulting in top-down increases in auditory understanding (*Franchak et al., 2016*; *Kirkorian et al., 2012*). Unfortunately, as eye movements were not recorded, we cannot establish the relationship between eye movements and synchrony in sensory cortex (*Alexander et al., 2017*). The growth of coordinated responses with age may also drive the stronger event models in these regions (*Figure 5*). The ISC_b analysis (*Figure 2*) found relatively small but significant shifts in the group-level responses in sensory regions, and auditory responses shifted slightly earlier in time with age (*Figure 6*). This indicates that the age-related changes in sensory regions are not only largely characterized by increasing synchrony across subjects, but also (to some extent) a shift from a more child-like to more adult-like timecourse of responses.

Theory of mind regions responsible for representing the mental states of others (*Gallagher et al., 2000*; *Saxe and Kanwisher, 2003*), including TPJ (*Samson et al., 2004*), develop immensely in the age range studied (*Sebastian et al., 2012*). We found very large age-related changes in the functional responses of TPJ (*Figure 2*). Older children, due to stronger schematic event representations, are

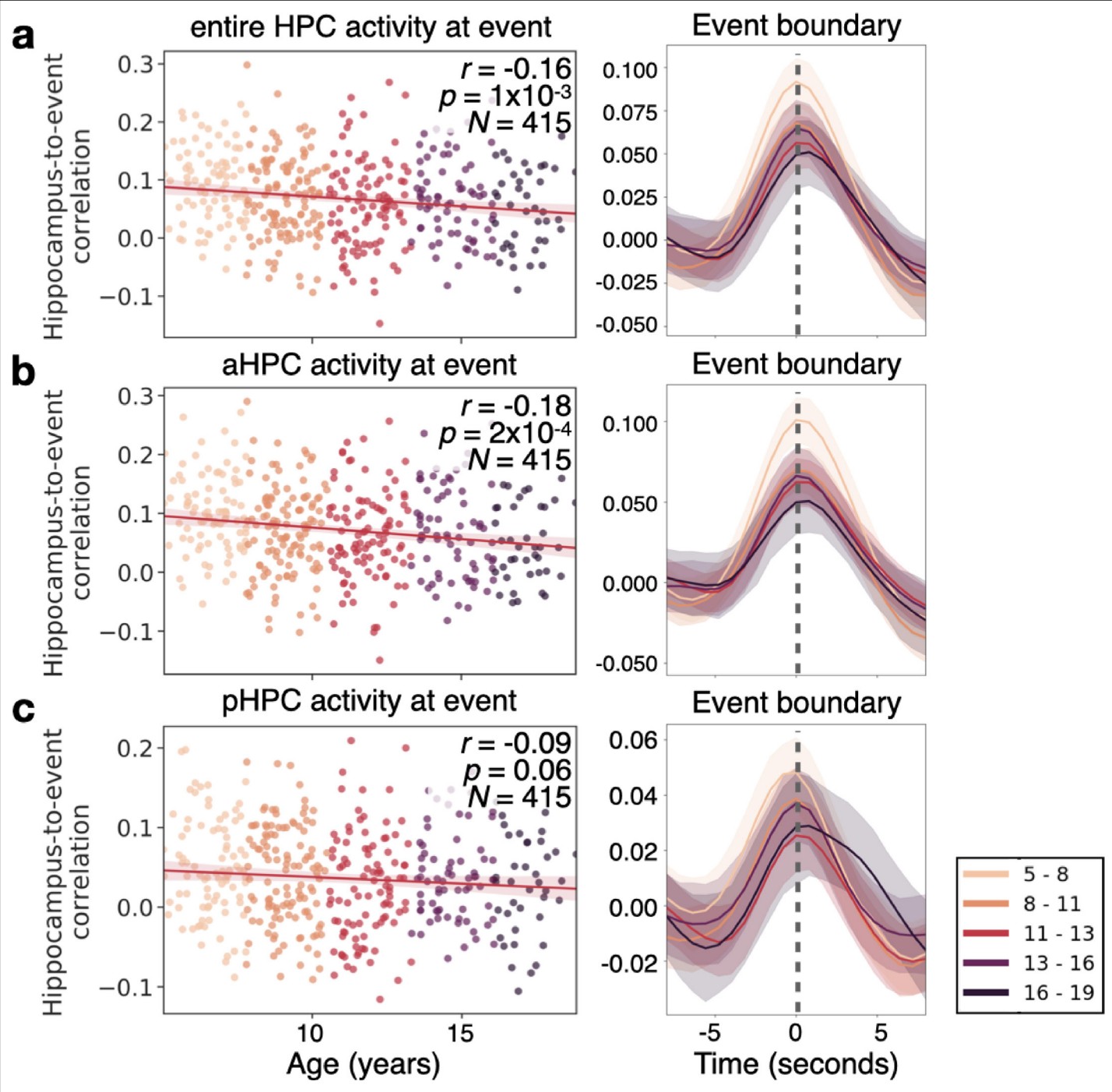

**Figure 7.** The hippocampus in children as young as 5 responds to event boundaries, an effect that decreases with age. (**a**) The event-boundary driven response decreases as a function of age in the hippocampus (HPC). This can be shown across individuals at the time point where hippocampus-to-event correlation is maximal (left), and throughout the time-course of the HPC's correlation with event boundaries. (**b**) The correlation between the HPC and event boundaries significantly decreases with age in the anterior HPC (aHPC), but in (**c**) posterior HPC (pHPC) there is no significant relationship between the boundary-driven signal and age. The shaded region surrounding each line represents the 95% confidence interval across all 415 subjects.

likely able to construct better internal models of social and emotional structure (*Dumontheil et al., 2010*; *Pavias et al., 2016*), allowing them to form more cohesive models of the world. Previous work found that responses in a network of regions, including TPJ, shift approximately 2 s earlier when 6–7 year-olds watch a movie for a second time (*Richardson et al., 2019*). In the right posterior-superior temporal sulcus (part of TPJ as defined by *Dufour et al., 2013*) and in the TPJ-adjacent left

postcentral sulcus, we found that neural representations in 16–19 year-olds can be as far as 20 s ahead of those for 5–8 year-olds on the *first* viewing of a movie clip. This reveals a protracted development in this region, likely due to continued growth in the ability to proactively represent the minds of others (*Dumontheil et al., 2010*; *Pavias et al., 2016*), or possibly due to a growth in attentional abilities in this region more generally (*Corbetta and Shulman, 2002*; *Downar et al., 2001*).

We also found age-related shifts in dlPFC (*Figure 2*) and increasing anticipation from ages 5–11 (*Figure 6*). This region, along with other regions showing age-related shifts such as the inferior temporal cortex and orbitofrontal cortex, are a part of a network responsible for processing conceptual knowledge about the world (*Binder et al., 2009*). The age-driven differences in story responses may therefore be partially due to changes in the understanding of the people, actions, and culture presented in the movie (*Binder et al., 2009*). The dlPFC is also critical for working memory, or the ability to keep recently learned information readily accessible (*Curtis and D'Esposito, 2003*). The dlPFC exhibits an improvement in event structure with age (*Figure 5*), which may be due to the ongoing development of working memory (*Kwon et al., 2002*), a cognitive function necessary to chunk incoming information into discrete events (*Clewett et al., 2019*). The changes in dlPFC may also be related to development of theory of mind as discussed above, as it overlaps with the theory of mind network identified by *Richardson et al., 2019*.

In the PMC event model fits worsened with age (*Figure 5*), and in the more posterior RSC portion, reliability also decreased with age (*Figure 1*). This finding contrasts with our hypothesis that schematic event representations should increase with age in regions such as the PMC. One possible reason for this discrepancy is that PMC performs additional processing due to its lack of strong schemas about observed events; *Keidel et al., 2018* found that the PMC (RSC) had significantly more activity in response to movies when the narrative topic was new than when it was a continuation of themes from the previous clip. Younger children may therefore more consistently rely on this region to understand a relatively more novel environment. RSC is also known to be responsible for processing visual scenes and is fully developed by age 7 (*Epstein and Baker, 2019*; *Meissner et al., 2019*). It is therefore possible that young children are more consistently reliant on spatial scene representations as they process an ongoing narrative. PMC has an intrinsically long timescale for information processing, one that emerges in infancy, which may thus support strong event models in children as young as five (*Hasson et al., 2015*; *Stephens et al., 2013*; *Yates et al., 2021*). However, we did find both age-related shifts in the pattern by which the narrative was represented bilaterally (*Figure 2*) and increased anticipation in the left PMC (*Figure 6*). These findings in the more dorsal portion of PMC are notable because there were no significant age-related differences in this region, suggesting that these shifts are not due to overall changes in attention or executive function. This result is in line with our hypothesis that increasing age is associated with more anticipation of higher level themes, and different ages likely process these themes with a different overall pattern of activity.

We found small, but significant changes in the timecourse of narrative processing in the medial prefrontal cortex (*Figure 2*). This region has been previously implicated in tracking schematic knowledge during movie-watching (*Baldassano et al., 2018*; *van Kesteren et al., 2010*), suggesting that schematic responses evolve across this age range. However, we surprisingly did not find a change in the strength of event representations with age. This may be due to an overall low level of ISC in this region, either due to idiosyncratic cognition, inconsistent neuroanatomy across subjects, or noisy signals in this region (*Figure 1—figure supplement 1*), making it challenging to study this region using the group-level approaches in this study. This result is not entirely unexpected given that previous studies have found low levels of ISC in this region (*Baldassano et al., 2020*), especially in developmental populations (*Lerner et al., 2019*; *Moraczewski et al., 2018*). Furthermore, as has been found previously, it may be that only a subregion of the medial prefrontal cortex represents events (*Baldassano et al., 2020*; *Brod et al., 2017*; *Masís-Obando et al., 2022*; *van Kesteren et al., 2010*).

The HPC was found to reliably respond to the boundaries between events (*Figure 7*). This result extends evidence that the HPC tracks event changes in adults to children as young as five (*Ben-Yakov and Henson, 2018*; *Baldassano et al., 2020*; *Reagh et al., 2020*). This boundary-triggered response decreased with age across our sample, which is surprising given that adolescents have stronger event models and event anticipation in most of the cortical regions with significant effects. The decrease in HPC responsiveness with age may be a signature of age-related increases in the reliance on schematic

representations of the environment which are stored in the cortex, thus decreasing the reliance on the HPC to encode newly encountered events (*Sekeres et al., 2018*). The HPC may thus have a more robust signal in young children who need to focus on encoding the specific details of events as they use them to extract commonalities across experiences (*Keresztes et al., 2018*; *Maril et al., 2010*). Likewise, the HPC has been shown to have increased activity during memory consolidation, and in response to novel events (*Nadel and Moscovitch, 1997*; *Stern et al., 1996*). It is possible that younger children found the events in the movie generally more novel, or needed to spend more effort consolidating them due to their impoverished schematic representations. However, if this was truly a novelty-related response we would have expected to see a greater age-related change in the pHPC which has been shown to be more responsive to novelty (*Stern et al., 1996*).

Interestingly, previous research in older adults also found a decrease in the HPC response to event boundaries (*Reagh et al., 2020*). However, in this previous research the decrease with age was seen in the pHPC (*Reagh et al., 2020*). Here, we find that only the response in the aHPC significantly decreases with age. This shift in age-related decline may be due to the developmental changes in the mnemonic roles along the long axis of the HPC (*Demaster and Ghetti, 2013*; *Langnes et al., 2019*). The decline in the response of the aHPC may also be related to the process of semanticization (*Sekeres et al., 2018*), whereas older ages may be experiencing age-related mnemonic, and thus perceptual declines (*Reagh et al., 2020*).

Jointly fitting an HMM across children of different ages was sensible because we found that the optimal event timescales in all regions with high model fits were similar between the Youngest and Oldest groups. This was not a foregone conclusion, because infants under one year of age segment ongoing narratives into far fewer events than adults (*Yates et al., 2021*) and behaviorally, we found small, but significant differences in segmentation behavior between adults and children, and children of different ages. However, at least neurally, children therefore develop enough narrative comprehension to segment the story in an adult-like manner by age five.

Although we largely attribute the better segmentation ability and prediction to adolescents' greater experience with the world in general, it is possible that our Oldest cohort also had more previous experience watching the specific stimulus used (a clip from "Despicable Me"), as it was first released when they were young children themselves. Alternatively, younger children may have had more recent exposure to the film because it was more age appropriate for them, and this may have elicited a retrieval induced increase in HPC activity. Future research should collect data on prior stimulus exposure or aim to use stimulus-naive subjects for better control. Furthermore, we cannot distinguish whether the improvements that we see with age are due to the overall maturation of the brain or the increased life experience that underlies schematic knowledge. Additionally, although we chose to use this dataset because it is one of the few large databases of movie-watching in a developmental population, the subjects recruited for this study had a high prevalence of psychopathology and limited socioeconomic diversity. Future work should generalize our findings to more diverse and representative subject samples. Finally, our analyses require stimuli that exhibit multiple transitions between events that are at least 30 s. We were therefore unable to make use of the other movie available in this dataset or data from the short video clips used in other developmental datasets (*Alexander et al., 2017*; *Richardson et al., 2019*). Future studies should seek to generalize these results to stimuli with greater content diversity, and duration.

Our results reveal that brain responses to a narrative movie do not simply become more synchronized across children as they age, but in fact change their dynamics and timing to become more adult-like. Brain activity patterns in many regions become more cleanly structured into discrete, consistent events, and the timing of these events shifts to anticipate upcoming content in the movie. We also found that hippocampal responses to event boundaries are present in children as young as 5 years old, and that these event-boundary responses are in fact larger in children than in adolescents. This study provides the groundwork for assessing how children acquire schematic knowledge about the world and ultimately deploy that knowledge at the appropriate time.

# Materials and methods

## Participants

The data in these analyses was downloaded from the Child Mind Institute's Health Brain Network (HBN) project (*Alexander et al., 2017*) when 1758 MRI datasets were available (data release 6). All participants provided written consent or assent, and consent was obtained from the parents or legal guardians for participants younger than 18 years. The goal of the HBN project was to collect data from subjects with heterogeneous developmental psychopathologies. As such, although the demographics and prevalence of psychopathology may not be representative of the general population, this is one of the largest developmental datasets collected while children viewed natural stimuli, and it therefore allows us to do the novel analyses outlined below. The HBN project was approved by the Chesapeake Institutional Review Board. No task functional magnetic resonance imaging (fMRI) data was available for 343 subjects. Of the remaining subjects, 803 subjects were rejected because they did not contain recordings for one of the two possible movie stimuli, they lacked one of the two possible fieldmap scans, or one of the two movies had a different number of time samples than was expected if the subject watched the entire movie. Of the remaining subjects, 183 were rejected due to unacceptably poor T1 scans, as assessed by four independent raters. Two additional subjects were eliminated as the field of view of the acquisition did not cover the whole brain, one subject was eliminated due to a damaged T1 file, and one subject was eliminated due to median framewise displacement greater than three standard deviations above the mean. Four hundred and twenty six subjects remained after these eliminations. To allow for a sufficient number of subjects (N=40) in each age group spanning 2.76 years, with an equivalent ratio of males to females in each age group, subjects older than 19 were not used in further analyses (N=11). Most analyses were computed on gender-matched (22 males) samples of the Youngest (5.04–7.8 years, rounded in displays to 5–8 years) and Oldest (16.10–18.87 years, rounded in displays to 16–19 years) subjects, although illustrative analyses were also computed on the intermediate age groups. The subjects were not evenly distributed across the age groups (*Figure 1—figure supplement 3*). Many different groups of 40 subjects could therefore be drawn from the Youngest group (N=87, 47 males). Since the results change slightly depending on which group of 40 was selected, we conducted our analyses on the average of five random subsamples of each age group.

## MRI data collection

Magnetic Resonance Imaging (MRI) data used in this study were collected at both the HBN Rutgers University Brain Imaging Center site on a Siemens Trio Tim 3T scanner and the HBN CitiGroup Cornell Brain Imaging Center site on a Siemens Prisma 3T scanner. The scan parameters at both sites were identical: TR = 800ms, TE = 30ms, # slices = 60, flip angle = 31°, # volumes = 750, voxel size = 2.4 mm. Complete information regarding the scan parameters used for the Healthy Brain Network project can be found at: http://fcon_1000.projects.nitrc.org/indi/cmi_healthy_brain_network/MRI%20Protocol.html.

## Stimulus and event rating

fMRI data were collected while participants viewed a 10 min clip from the movie *Despicable Me* (01:02:09–01:12:09; presentation details available at http://fcon_1000.projects.nitrc.org/indi/cmi_healthy_brain_network/MRI%20Protocol.html). To obtain the timing of event boundaries, a separate group of 21 adult raters (6 males) watched the clip and were given the following instructions: "The movie clip can be divided into meaningful segments. Record the times (in seconds) denoting when you feel like a meaningful segment has ended. Pause the clip at the end of the segment, write down the time in the attached spreadsheet, and provide a short, descriptive title. Try to do this when you watch the clip for the first time." No one reported watching the clip more than once. Event boundaries were determined as time points where over half of the group agreed an event had occurred and are displayed for reference at the top of *Figure 3*. On average, raters found 14.4+/-6.5 events in the movie (minimum = 4, maximum = 30).

The perceived timing of event boundaries according to participants within the age range that fMRI was recorded (5–19 years) was assessed online via the Gorilla platform (https://gorilla.sc; *Anwyl-Irvine et al., 2020*). The online experiment involved a brief training to ensure comprehension of the task. Participants first watched a different clip from Despicable Me (03:25 to 04:47 in the full

movie). Participants were notified after the first event boundary (04:10 in the full movie) and asked to identify the next event boundary (04:37 in the full movie). Participants were given three attempts to correctly identify the second boundary. If a participant successfully identified the second boundary, they watched the same clip that was used during fMRI scanning and identified boundaries in that clip at their discretion. The children's data were first analyzed in aggregate, and then analyzed after splitting based on the median age (12.25 years).

The online task was also administered to 25 adults (9 male, 19–56 years) recruited via Prolific (https://www.prolific.co). The data from two adults were not analyzed as their ages overlapped with the age range of the fMRI participants (they were under 19 years). Data from 66 participants (39 male, 7–18 years) within the age range of the fMRI participants were recruited via word of mouth and Facebook. The data from eight subjects were eliminated as their median event duration was less than one second. All online experimental procedures were approved by the Columbia University IRB (protocol number AAAS0252, for adults, and AAAT8550, for children), and the data is available online: https://github.com/samsydco/HBN.

## Anatomical data preprocessing

The results included in this manuscript come from preprocessing performed using fMRIPprep 1.1.4 (*Esteban et al., 2018*; RRID:SCR_016216), based on Nipype 1.1.1 (*Gorgolewski et al., 2011*; RRID:SCR_002502). The T1-weighted (T1w) image was corrected for intensity non-uniformity (INU) using `N4BiasFieldCorrection` (*Tustison et al., 2010*; ANTs 2.2.0; RRID:SCR_004757), and is referred to as 'T1w-reference' throughout the workflow description. The T1w-reference was then skull-stripped using `antsBrainExtraction.sh` (ANTs 2.2.0), using OASIS as a target template. Brain surfaces were reconstructed using `recon-all` (*Dale et al., 1999*; FreeSurfer 6.0.1,RRID:SCR_001847), and the brain mask estimated was refined with a custom variation of the method used to reconcile the ANTs-derived and the FreeSurfer-derived segmentations of the cortical gray-matter in Mindboggle (*Klein et al., 2017*; RRID:SCR_002438). Spatial normalization to the ICBM 152 Nonlinear Asymmetrical template version 2009c (*Fonov et al., 2009*; MNI152NLin2009cAsym; RRID:SCR_008796) was performed through nonlinear registration with `antsRegistration` (ANTs 2.2.0, RRID:SCR_004757), using brain-extracted versions of both the T1w volume and the template. Brain tissue segmentation of cerebrospinal fluid (CSF), white-matter (WM) and gray-matter (GM) was performed on the brain-extracted T1w using `fast` (*Zhang et al., 2001*; FSL 5.0.9, RRID:SCR_002823).

## Functional data preprocessing

A reference volume and its skull-stripped version were generated using custom fMRIprep methodology. For the cortical results, a B0-nonuniformity map (or fieldmap) was estimated based on two echo-planar imaging (EPI) references with opposing phase-encoding directions, with `3dQwarp` (*Cox, 1996*; AFNI 20160207). Based on the estimated susceptibility distortion, a corrected EPI reference was calculated for a more accurate co-registration with the anatomical reference. For the hippocampal results, "Fieldmap-less" distortion correction was performed by co-registering the functional image to the same-subject T1w image with intensity inverted, (*Huntenburg, 2014*; *Wang et al., 2017*) constrained with an average fieldmap template (*Treiber et al., 2016*), implemented with antsRegistration (ANTs).

Head-motion parameters with respect to the BOLD reference (transformation matrices, and six corresponding rotation and translation parameters) were estimated before any spatiotemporal filtering using `mcflirt` (*Jenkinson et al., 2002*; FSL 5.0.9). The BOLD time-series was resampled into its original, native space by applying a single, composite transform to correct for head-motion and susceptibility distortions. This resampled BOLD time-series will be referred to as 'preprocessed BOLD'. The BOLD reference was then co-registered to the T1w reference using `bbregister` (FreeSurfer) which implements boundary-based registration (*Greve and Fischl, 2009*). To account for distortions remaining in the BOLD reference, co-registration was configured with nine degrees of freedom. The BOLD time-series was resampled to the 'fsaverage6' space for the cortical analyses, and to the 'MNI152NLin2009cAsym' space for the hippocampal analyses. Confound time-series were calculated based on the preprocessed BOLD. Framewise displacement (FD) was calculated using the Nipype implementation (*Power et al., 2013*). Global signals were extracted within the CSF and the WM masks. Eight discrete cosine filters were extracted with 128 s cut-off. These nuisance regressors,

as well as the head-motion estimates, were placed in a confounds file and subsequently regressed out of the BOLD time series separately for the cortex and the hippocampus using custom python scripts. All resamplings can be performed with a single interpolation step by composing all the pertinent transformations (i.e. head-motion transform matrices, susceptibility distortion correction, and co-registration to anatomical and template spaces). Gridded (volumetric) resamplings were performed using `antsApplyTransforms` (ANTs), configured with Lanczos interpolation to minimize the smoothing effects of other kernels. Non-gridded (surface) resamplings were performed using `mri_vol2surf` (FreeSurfer).

Many of the internal operations of fMRIPrep use Nilearn (version 0.4.2, *Abraham et al., 2014*, RRID:SCR_001362), mostly within the functional processing workflow. (For more details of the pipeline, see https://fmriprep.readthedocs.io/en/latest/workflows.html "FMRIPrep's documentation".) After these preprocessing steps were taken, the cortical surface was parcellated into the 100 parcel, seven network parcellation from *Schaefer et al., 2018*.

## Defining temporoparietal junction

To better understand how our findings relate to the previous literature we defined the portion of the temporoparietal junction (TPJ) responsible for theory of mind processes based on its definition in *Dufour et al., 2013*. We projected the definition of the TPJ from this paper, originally in voxel space, onto the cortical surface, and parcels in which over two-thirds of their vertices were within the boundary of the TPJ were labeled as members of this region.

## Inter-subject correlation (ISC) calculations

After preprocessing, split-half inter-subject correlation (shISC) was calculated by splitting the group within which ISC will be calculated, into two equal halves, averaging each halves' timecourse, and then correlating the two average timecourses. We chose to use shISC since we found that it could produce ISC estimates for a large group much more quickly than using pairwise ISC (pwISC) and leave-one-out ISC (looISC). Using a model similar to *Nastase et al., 2019*, it can be shown that, in expectation, pwISC and looISC can be calculated from shISC if the sample size is known (see Appendix 2). shISC was calculated for each vertex on the cortical surface, and then averaged within each parcel.

The overall level of ISC among the participants, irrespective of age, was calculated in a randomized mixture of the Youngest and Oldest subjects (*Figure 1—figure supplement 1*). This calculation can be viewed as a quality assurance step because if there is no baseline level of ISC, it is possible that the movie was not engaging to subjects, or that the sample is dominated by movement artifacts (*Cohen et al., 2017*; *Vanderwal et al., 2020*). After establishing the level of the overall group ISC, ISC was calculated within the Youngest and Oldest groups and compared between the groups, following the methods used in previous research (*Cantlon and Li, 2013*; *Moraczewski et al., 2018*). To ensure that any regions in which ISC increased with age were not driven by an increase in the level of noise in the Youngest group, we calculated the correlation between each individual's median framewise displacement and their looISC with the other subjects for the Youngest group in parcels where ISC was found to increase with age.

In addition to measuring within-group ISC, we sought to compare the similarity between age groups (while controlling for within-group consistency). In order to assess a potential difference in stimulus response timecourses between the groups, while accounting for the relative level of within group correlation, we introduce a measure named between-group ISC (ISC$_b$). Previous researchers have measured the timecourse correlations between subjects in two different groups (*Cantlon and Li, 2013*; *Hasson et al., 2009*; *Moraczewski et al., 2018*; *Nastase et al., 2019*), but this measure mixes information about across-group differences and within-group consistency. For example, if two groups are identical, we would expect the between-group correlation to be the same as the within-group correlation. ISC$_b$ disentangles these effects, producing an estimate of the correlation between the average timecourses of two groups in the limit of infinite data (in which the group-average timecourses are measured without noise). Mathematically, ISC$_b$ is computed as the correlation between two different groups divided by the geometric mean of the correlation within each group. The denominator means that ISC$_b$ accounts for differences in ISC magnitude between the groups (see Appendix 1 for mathematical derivation). ISC$_b$ was calculated between the Youngest and Oldest groups, using the shISC formulation (Appendix 1).

## Analysis of event segmentation behavior

Event segmentation behavioral data from each individual was first convolved with a canonical hemo-dynamic response function (HRF, one parameter gamma model from AFNI's 3dDeconvolve) at each timepoint (*Lee et al., 2021*). This data was compared between children, adults, and the younger and older children as determined by a median split based on age. Data was compared within and between groups by splitting each group in half, and averaging the data across raters within a split, thus computing shISCs where appropriate. This procedure was done on five random splits of the data. The $ISC_b$ measure was used to compare response timecourses between the groups. These values were computed across five random splits of the data and results were averaged across the splits. All significance values were computed on 10,000 random permutations of the data.

## Calculating the number of events in each age group

To determine whether the Oldest and Youngest groups perceived the same number of events in the movie, following *Baldassano et al., 2020*, we define an event as a stable spatial pattern of neural activity within a parcel. In each age group, and each parcel, event patterns and the number of events are chosen using a hidden markov model (HMM). To find the best fitting model, HMMs that used between 2 and 50 events (approximately logarithmically spaced) were trained and tested using five-fold cross-validation by first averaging the brain activity of four-fifths of the subjects in an age group, and then testing on the average brain activity of the held-out subjects. The number of events in an age group were defined as the average of the number of events that maximized the log likelihood of the HMM (model fit) in the held out subjects in each of the five cross-validated folds.

Although most cortical regions are expected to exhibit event structure in response to a movie stimulus, some regions will not have activity that can be well modeled by a model of events. The difference in model fit between a baseline HMM in which two events were fit and the HMM with the number of events yielding the best fit was used as a measure of the extent to which event structure was successfully detected in this region (*Figure 5—figure supplement 1*). Regions that do not show substantial improvement in model fit over the two-event model either have preferred event lengths on the scale of 300 s or longer (since the stimulus was 600 s long) or do not have event-structured responses. In 30 parcels both the Youngest and Oldest groups had a model fit difference less than 0.002 and were therefore not considered in future analyses (*Figure 5—figure supplement 1*). In 40 parcels, the Youngest group had a model fit difference less than 0.002 and in 33 parcels the Oldest group had a model fit difference less than 0.002. In the remaining 71 parcels with a good model fit in at least one group, there was no significant difference in the number of events between the groups (*Figure 5—figure supplement 2*; see Statistical Significance section for permutation test details).

## Comparison of event boundaries in brain regions to annotations

Following the methods of *Lee et al., 2021*, we jointly fit an HMM in each parcel to the Youngest and Oldest groups simultaneously. This joint-fitting procedure assumes that both groups share the same number of events and brain pattern in each event. Both the degree of model fit and the timing of these events can vary between the groups. The number of events and event patterns were determined by finding the maximum log likelihood of the model-fit on the average of five held-out folds of the data from all five age groups (including ages in between the Youngest and Oldest). This fitting procedure is similar to the case where HMMs are fit separately in each age group, except in this case, the log likelihood of the model-fit is averaged over all age groups in addition to all training folds.

We next evaluated the correspondence between the HMM-derived event boundaries and boundaries from age-matched children. To determine where event transitions occur in the model, we computed the derivative of the expected event label over time for each age group and parcel. The behavioral boundaries were derived from the number of boundary annotations at each timepoint convolved with a canonical HRF, as was done in the Analysis of event segmentation behavior (*Lee et al., 2021*). We related the two timecourses using Pearson correlation (*Figure 5—figure supplement 2*).

### Determining the model fit in each age group

The log likelihood, or model-fit, of the best-fitting jointly-fit HMM was compared between the Oldest and Youngest groups. For illustration of the change in log likelihood with age, the average log likelihood was also computed on similarly sized folds of the age groups in between the Youngest and Oldest groups.

### Measuring changes in event timing across age

Following the procedure from *Lee et al., 2021*, using the jointly-fit HMMs described in the previous section, the probability that each fold of the held-out data was in any of the events was determined for each age group in each parcel. Averaging these probabilities across all five-folds yielded a time-point by number of events matrix. We then computed (separately for each group) the expected value of the event present at each timepoint. Summing over these expectation values produced a value describing the amount of time that this region spent in earlier versus later events, with larger numbers indicating that more time was spent in later events. If the sum of these expectation values is greater in one age group than in the other, this indicates that this age group represents upcoming events before the other age group.

In all parcels that had a significant change in event timing between age groups, we compared the timing of the event boundaries identified by the HMM to the event boundary timecourse obtained behaviorally from raters. Following methods used previously, the number of boundary annotations from the child raters at each timepoint during the movie were convolved with a canonical HRF (*Lee et al., 2021*). To obtain a measure of when the brain is switching between events, we took the derivative of the expected value of the event that the HMM assigned to each timepoint (following methods from *Lee et al., 2021*). To determine the timing of the alignment between these two signals, we cross-correlated the HMM-derived boundary timecourse with the behavioral event boundary timecourse from children, in aggregate. To precisely estimate the optimal lag, we fit a quadratic function to the maximum correlation lag and its two neighboring lags, and found the location of the peak of this quadratic fit. We computed the maximum-correlation lag separately in the Oldest and Youngest groups to determine whether there is a significant difference in the relationship between the brain's event boundaries and the behavioral event boundaries.

### Statistical significance

To test for a significant difference between the Youngest and Oldest groups in all of the above analyses, Age permuted values were computed for each parcel for the difference in ISC magnitude (absolute value of ISC difference), $ISC_b$, number of events in age group specific HMMs, difference in model fit (maximum log likelihood) for jointly-fit HMMs, difference in event timing for jointly-fit HMMs, and difference the timing of the jointly-fit HMMs events with behavior. All age permuted groups had the same gender distribution as the true sample. For each parcel, the calculation of the real (non-permuted) value, as well as all permuted values were calculated in five random subsamples of each age group (Youngest and Oldest) since random subsampling generates slightly different results. Both the real and permuted values were then averaged across all five subsamples to calculate both the values reported here and their significance. The permuted group assignment in each permutation was therefore maintained across the five subsamples.

If in any parcel, for any test, the number of the permuted values greater than (or less than, in the case of $ISC_b$) the true value was less than 5% in any of the five random subsamples after 100 permutations, 1000 additional permutations were run. A total of 1000 additional permutations were computed until every parcel had at least one permuted value greater than (or less than) the value in the original data, or 6000 permutations had been computed (whichever came first). If after 6000 permutations, there were still no permuted values greater than (or less than) the value in the original data, the original data was included as a permuted value for significance tests.

All HMMs were initially trained on the original (non age-permuted) training data (four-fifths of the full dataset). To compute permuted differences of the number of events, age-specific HMMs were tested on age-permuted testing data for all numbers of events. The jointly-fit HMMs were tested on permuted data for only the best fitting number of events, determined from the original data. In all statistical tests, the false discovery rate (FDR) of parcel differences was controlled for by setting the expected proportion of false positives to.05 (*Benjamini and Hochberg, 1995*). The code for all the included analyses is available at: https://github.com/samsydco/HBN.

## Hippocampal segmentation and event rating correlation

The hippocampus was defined using the Freesurfer segmentation generated using fMRIPrep 1.5.6 (*Esteban et al., 2018*). Consistent with previous studies, the anterior hippocampus (aHPC) was defined as voxels anterior to y = –21 mm in MNI space, and the posterior hippocampus (pHPC) was defined as voxels including and posterior to y = –21 mm in MNI space (*Poppenk et al., 2013*; *Zeidman et al., 2015*). For each subject, voxel-wise activity was averaged within the entire hippocampus, the aHPC, and the pHPC. Each subject's hippocampal timecourses was then correlated with an event boundary timecourse obtained behaviorally from age-matched children. Similarly to the behavioral timecourse related to the HMM, here, the event boundary timecourse was computed by convolving the number of boundary annotations at each timepoint during the movie with a canonical HRF (*Lee et al., 2021*). The hippocampal timecourses and the event boundary timecourse were correlated at differences in delay of up to 10 s. These lagged correlations were computed because previous research has shown that the hippocampal response to event boundaries can follow event boundaries by several seconds (*Baldassano et al., 2020*; *Ben-Yakov and Henson, 2018*; *DuBrow and Davachi, 2016*; *Reagh et al., 2020*). The correlation values for each hippocampal region, temporal lag, and subject were then grouped according to the age of the subject. The Pearson's correlation at lag 0 between age and the hippocampus-to-event boundary correlation was assessed across all subjects in the entire hippocampus as well as in all hippocampal subregions.

Previous research has shown that the volume of the hippocampus may decrease with age (*Gogtay et al., 2006*). The volume of the hippocampus and its subregions was therefore measured as the number of voxels that the Freesurfer parcellation allotted to the hippocampus. The aHPC volume was the number of voxels anterior to y = –21 mm in MNI space, and the pHPC volume was the number of voxels including and posterior to y = –21 mm in MNI space. The volume of the entire hippocampus and aHPC correlated with subject age (entire HPC: $r=-0.1$, $p=0.03$, aHPC: $r=-0.1$, $p=0.03$) while the volume of the pHPC did not ($r=-0.05$, $p=0.3$). However, in none of the regions was the age-matched event boundary response significantly correlated with the change in volume (entire HPC, $r=0.01$, $p=0.9$, aHPC, $r=-0.01$, $p=0.9$, pHPC, $r=-0.03$, $p=0.6$).

## Acknowledgements

We thank Andrew Africk for providing funding support for S.C., Michael Chow for helping to formulate the math for ISCb, and Nora Newcombe, Paul A Bloom, Rolando Masis-Obando, and Vishnu "Deepu" Murty for their edits and suggestions.

## Additional information

### Funding

| Funder | Grant reference number | Author |
|--------|------------------------|--------|
| Andrew Africk | | Samantha S Cohen |

The funders had no role in study design, data collection and interpretation, or the decision to submit the work for publication.

### Author contributions

Samantha S Cohen, Conceptualization, Data curation, Formal analysis, Methodology, Software, Visualization, Writing - original draft, Writing - review and editing; Nim Tottenham, Conceptualization, Investigation, Methodology, Project administration, Resources, Supervision, Writing - review and editing; Christopher Baldassano, Conceptualization, Funding acquisition, Methodology, Resources, Supervision, Writing - review and editing

### Author ORCIDs

Samantha S Cohen http://orcid.org/0000-0003-3007-5372
Christopher Baldassano http://orcid.org/0000-0003-3540-5019

## Ethics

Human subjects: Informed consent, and consent to publish, was obtained from all subjects 18 years and older. Consent was obtained from the parents or legal guardians for participants younger than 18 years. The neuroimaging portion of the study was approved by the Chesapeake Institutional Review Board (https://www.chesapeakeirb.com/). The behavioral experimental procedures were approved by the Columbia University IRB (protocol number AAAS0252, for adult data, and AAAT8550, for child data).

## Decision letter and Author response

Decision letter https://doi.org/10.7554/eLife.69430.sa1
Author response https://doi.org/10.7554/eLife.69430.sa2

## Additional files

### Supplementary files

• Transparent reporting form

### Data availability

All neuroimaging data is available at: http://fcon_1000.projects.nitrc.org/indi/cmi_healthy_brain_network/sharing_neuro.html, and all behavioral data is available at: https://github.com/samsydco/HBN, (copy archived at swh:1:rev:278127d07b721c73679c11d0d1836631df778323).

The following dataset was generated:

| Author(s) | Year | Dataset title | Dataset URL | Database and Identifier |
| --- | --- | --- | --- | --- |
| Cohen S | 2022 | HBN | https://github.com/samsydco/HBN | GitHub, 278127d |

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

## Appendix 1

For across-group analyses, we want to calculate the ISC between two groups of subjects, with two different sets of shared responses, $g_1$ and $g_2$, and different levels of variance, $\epsilon_i$ and $\epsilon_j$. To simplify the notation, we define $g_1^T g_1 = a_1$, $g_2^T g_2 = a_2$, $g_1^T g_2 = a_B$, $\mathbb{E}[\epsilon_i^T \epsilon_i] = b_1$ and $\mathbb{E}[\epsilon_j^T \epsilon_j] = b_2$. We would like to estimate the correlation between $g_1$ and $g_2$, regardless of the difference in signal-to-noise levels between the two groups. That is, our goal is to compute:

$$ISC_b = \frac{a_B}{\sqrt{a_1}\sqrt{a_2}}$$

If we know the ISCs for each group, from **equation 1**:

$$shISC_1 = \frac{N^2 a_1}{N^2 a_1 + 4N b_1}$$

and

$$shISC_2 = \frac{N^2 a_2}{N^2 a_2 + 4N b_2},$$

where $N$ is the number of subjects in each group, and the ISC between the two groups is:

$$shISC_B = \frac{N^2 a_B}{\sqrt{N^2 a_1 + 4N b_1}\sqrt{N^2 a_2 + 4N b_2}}.$$

Dividing $shISC_B$ by the geometric mean of $shISC_1$ and $shISC_2$ allows us to calculate the between-group ISC without the $b$'s:

$$ISC_b = \frac{shISC_B}{\sqrt{shISC_1}\sqrt{shISC_2}}$$
$$= \frac{a_B}{\sqrt{a_1}\sqrt{a_2}}.$$

## Appendix 2

We start by assuming that the response timecourse for every subject consists of a response that is shared across subjects, $g$, and a subject-specific response, $\epsilon$. The response timecourse for subject can therefore be represented as $s_i = g + \epsilon_i$. For simplicity, all subject timecourses, $s$, have zero mean, we define $a$ as the norm of $g$ ($g^T g = a$), and we define $\mathbb{E}[\epsilon_i^T \epsilon_i] = b$. If we have N subjects in total, the split half ISC ($shISC$) is:

$$
\begin{aligned}
shISC \quad &= \mathbb{E}\left[\rho\left(\sum_{i=1}^{N/2} s_i, \sum_{j=N/2+1}^{N} s_j\right)\right] \\
&= \mathbb{E}\left[\rho\left(\frac{N}{2}g + \sum_{i=1}^{N/2} \epsilon_i, \frac{N}{2}g + \sum_{j=N/2+1}^{N} \epsilon_j\right)\right] \\
&= \frac{\frac{N^2}{4}g^T g}{\sqrt{\frac{N^2}{4}g^T g + \mathbb{E}[\sum_{i=1}^{N/2}\epsilon_i^T\epsilon_i]}\sqrt{\frac{N^2}{4}g^T g + \mathbb{E}[\sum_{j=N/2+1}^{N}\epsilon_j^T\epsilon_j]}} \\
&= \frac{N^2 a}{N^2 a + 2Nb} \\
&= \frac{Nf}{Nf+2}
\end{aligned}
\tag{1}
$$

where $f = a/b$, or the signal to noise ratio (the ratio of norms between $g$ and $\epsilon_i$).
If we rearrange the variables in **Equation 1**, we find that:

$$
f \quad = \frac{2 * shISC}{-N * shISC - N}.
\tag{2}
$$

We can thus calculate $pwISC$ and $looISC$ from $shISC$ as they relate to $f$ as follows. For $pwISC$ between subjects and $j$:

$$
\begin{aligned}
pwISC \quad &= \mathbb{E}[\rho(s_i, s_j)] \\
&= \mathbb{E}[\rho(g + \epsilon_i, g + \epsilon_j)] \\
&= \mathbb{E}\left[\frac{(g+\epsilon_i)^T(g+\epsilon_j)}{\sqrt{(g+\epsilon_i)^T(g+\epsilon_i)}\sqrt{(g+\epsilon_j)^T(g+\epsilon_j)}}\right] \\
&= \frac{g^T g}{\sqrt{g^T g + \mathbb{E}[\epsilon_i^T\epsilon_i]}\sqrt{g^T g + \mathbb{E}[\epsilon_j^T\epsilon_j]}} \\
&= \frac{a}{a+b} \\
&= \frac{f}{f+1},
\end{aligned}
$$

and for $looISC$, between one left-out subject, and all others:

$$
\begin{aligned}
looISC \quad &= \mathbb{E}\left[\rho\left(s_1, \sum_{j=2}^{N} s_j\right)\right] \\
&= \mathbb{E}\left[\rho\left(g + \epsilon_1, (N-1)g + \sum_{j=2}^{N} \epsilon_j\right)\right] \\
&= \frac{(N-1)a}{\sqrt{a+b}\sqrt{(N-1)^2 a + (N-1)b}} \\
&= \frac{(N-1)f}{\sqrt{f+1}\sqrt{(N-1)^2 f + (N-1)}} \\
&= \frac{\sqrt{N-1}f}{\sqrt{f+1}\sqrt{(N-1)f+1}}.
\end{aligned}
$$

Thus, for example, if there are 40 subjects, and $shISC$ is 0.5, the estimated $pwISC = 0.05$ and the estimated $looISC = 0.18$.

