## [Editor Report]

Cohen et al., present analyses of a large publicly available set of neuroimaging data from children and adolescents watching an animated video, and is likely to be of interest to neuroscientists interested in methods for analyzing naturalistic neuroimaging data, or those interested in the development of narrative processing in the brain. The methodological approach developed here is a valuable addition to the repertoire of developmental neuroscience.

---

## [Decision Letter]

**Decision letter after peer review:**

Thank you for submitting your article "Developmental changes in story-evoked responses in the neocortex and hippocampus" for consideration by *eLife*. Your article has been reviewed by 3 peer reviewers, including Peter Kok as Reviewing Editor and Reviewer #1, and the evaluation has been overseen by Floris de Lange as the Senior Editor. The following individuals involved in review of your submission have agreed to reveal their identity: Rebecca Saxe (with help from Frederik Kamps) (Reviewer #2); Zachariah M Reagh (Reviewer #3).

Essential revisions:

1. It needs to be carefully ruled out that (some of) the results can be explained by lower SNR in the younger age groups. See the reviewers' recommendations for authors for detailed suggestions.

2. It is not clear which hypotheses were tested, supported and refuted in the current study. We suggest you either improve the theoretical framework of the study, and clearly describe the hypotheses tested, or reframe the paper as an exploratory or methods paper. The reviewers provide some suggestions for how you might go about this.

3. It would greatly strengthen confidence in the current findings, which seem quite exploratory rather than hypothesis-driven, if they could be replicated in an independent data set. There are multiple public datasets of children watching videos, could any of these be leveraged to replicate the current findings?

4. The boundary ratings are obtained from adults, and may therefore not accurately reflect those of younger participants. It would strengthen the paper if boundary ratings could be obtained across the age range studied here.

5. How can we be sure developmental changes in the timing of event transitions reflect increasing "anticipation" in the older group, rather than lagging processes in the younger group? And are anticipatory shifts as long as 20s cognitively plausible?

6. Given the relatively uncontrolled nature of the stimuli, it is challenging to know exactly what video content drove responses for any given region. In addition to neural variance, differential eye and head movements induced by move content may also explain variance in the data. This should be acknowledged and discussed.

7. There is a relatively loose use of anatomy in places, e.g. regarding the TPJ. This should be either improved or acknowledged.

We encourage you to refer to the individual reviews for detailed recommendations on how to address these points.

*Reviewer #1 (Recommendations for the authors):*

– Could increased similarity in participants' eye movements (i.e. the way they overtly sample the videos) with age explain some of the effects reported here?

– Could there be a difference in head motion between age groups? Importantly, head motion might affect some regions (e.g. lateral ones) more than others (medial ones).

– Do the authors think that any of their effects are due to general development of the cortex rather than narrative comprehension? That is, might some of their effects also be found when participants watch scrambled videos without structure or meaning?

– Figure 6, right panels: can you make the different age groups more easily distinguishable?

– Seimen's -> Siemens

*Reviewer #2 (Recommendations for the authors):*

Major 1. The authors should significantly reframe the paper. There are many ways this could be done. One possibility is to hone attention toward a specific network most closely associated with "narrative" processing based on similar adult work, or which shows the strongest correlations to behaviorally-determined event boundaries. In this case, the authors should take care to provide clear definitions of the various terms in the manuscript, and clear hypotheses and alternative hypotheses. A second possibility is that the paper could be reframed in terms of the methodological approach (i.e., using HMM to explore event structure in naturalistic video data, which may have particular promise in pediatric datasets).

Major 2. Specific suggestions for addressing the concern that key effects are driving by lower data quality in younger children:

The authors should show within-group ISC across the cortex for young and old groups separately, not just the direct comparison of the two groups. Beyond confirming that the data in young children look reasonable, this analysis would reveal which regions actually show reliable within-group ISC (especially in the young group alone), and which do not. For example, it is possible that only occipital parcels show strong within-group ISC (as suggested by Supplemental Figure 1); if so, then it is not surprising to find no difference in within-group ISC between older and younger groups in regions beyond the occipital cortex, since reliable video responses could not really be detected there.

For the new between-group ISC measure (Figure 2), despite normalizing by within group ISC, this result could still be driven by lower data quality in younger kids than older kids. The strongest test of a qualitative developmental change in video responses would be to ask whether "young-young group" ISC is greater than "young-old group" ISC. If young children predict other young children better than older adolescents, then there is a reliable signal in the young group that cannot be explained by lower data quality in the younger sample than the older one. The alternative is that any comparison between a good dataset and a bad dataset will yield lower between group ISC than expected based on within-group ISC (i.e., driven by the older group only, by virtue of the higher data quality in that group).

The authors should also show HMM fits across the cortex, particularly in the young group alone. The comparison of model-fit-difference between young and old subjects in Supplemental Figure 2 is encouraging, although it appears that the young and old data are presented on slightly different scales (x versus y axis), and that more parcels did not show significant model fits in young children than older adolescents (based on the number of data points below the box "cutoff" line on each axis; the authors should report this number for each group directly, rather than the combined measure across groups). Further, in Figure 4 (side panels), it seems that the optimal timescale for the youngest group in many regions (TPJ, V1, A1) barely beats out the longest timescale, again calling into question the quality of the young child data. Finally, it would be helpful to know which parcels are showing the best model fit, and which show only weak model fits, again in both young and old groups separately.

*Reviewer #3 (Recommendations for the authors):*

I do not have what I would consider to be serious issues with the manuscript. In general, I think this is very good science and will make a solid contribution. I do, however, have some concerns that would make for a stronger publication if addressed:

(1) The authors obtained independent behavioral boundary ratings from adults, and compared these ratings to the neural responses from the participants. It does not appear that any of the independent raters were in the actual age range of interest (5-19 years). I suggest obtaining independent behavioral boundary ratings from the same age group to determine whether neural responses track with the behavior of the same age group. I will note that this discrepancy is not, in my view, a huge issue for the paper either way. Rather, I think there are interesting and important questions pertaining to this issue, namely whether behavioral measures pertaining to event boundaries differ fundamentally across this age range, and whether this has any bearing on the neural measures obtained in this study. It would be an equally novel and important contribution to demonstrate that this either does or does not have major influences on the patterns of results one might observe. While I am hesitant to request further data collection and/or analyses, in this case, I think it is warranted and would strengthen the kinds of conclusions that can be drawn from these data.

(2) On page 11, the authors suggest that the decreased response reliability, anticipation, and model fit performance is possibly due to the PMC performing "additional processing in the absence of strong schemas… [and] younger children may therefore more consistently rely on this region to understand a relatively more novel environment." It is unclear to me why people would anticipate an unfamiliar scene transition. Furthermore, based on the authors' prior work, it is unclear why the PMC, if it is relatively sensitive to higher-order event schemas (Baldassano, Hasson, and Norman, 2018), should show *more* anticipation by younger children. I am afraid I do not understand the logic here, or how to reconcile it with other findings. Can the authors clarify this?

(3) A general concern when comparing neural signals across different age ranges is the issue of signal quality, which can differ markedly across groups. The authors have seemingly accounted for potential differences in SNR in several formulae in the appendices, but do not mention in the main text whether age-related differences in SNR were examined or accounted for. This may perhaps be deducible from the formulae the authors provide, but regardless, I think a more direct mention of this issue would be helpful.

---

## [Author Response]

Essential revisions:1. It needs to be carefully ruled out that (some of) the results can be explained by lower SNR in the younger age groups. See the reviewers' recommendations for authors for detailed suggestions.

Upon re-examining the difference in framewise displacement (a measure of noise in fMRI data), we have realized that there was one outlier subject with a median framewise displacement of over three standard deviations above the mean. To ensure that the excessive motion in this subject did not affect any of our results, we have re-run all of the analyses in the paper without this subject.

As a result of eliminating this subject (from the youngest group), we have found slight changes in our results. Figure 1 reflects that there is no longer a significant increase in ISC with age in the visual cortex, and there is no longer a significant decrease in ISC with age in the anterior portion of PMC (PCC portion):

We have therefore modified the discussion to reflect this where we write:

“In the PMC event model fits worsened with age (Figure 5), and in the more posterior RSC portion, reliability also decreased with age (Figure 1).”

Additionally, we now find small but significant changes in ISCb in the medial prefrontal cortex in Figure 2.

We now address this in the discussion, where we write:

“We found small, but significant changes in the timecourse of narrative processing in the medial prefrontal cortex (Figure 2). This region has been previously implicated in tracking schematic knowledge during video-watching (Baldassano et al., 2018; Van Kesteren et al., 2010), suggesting that schematic responses evolve across this age range.”

Figure 6a reflects that there is no longer a decrease in anticipation with age in the PMC. We now find a decrease in anticipation in the right ventral temporal cortex.

We interpret the new increase in anticipation with age in the PMC in the discussion where we write:

“However, we did find both age-related shifts in the pattern by which the narrative was represented bilaterally (Figure 2) and increased anticipation in the left PMC (Figure 6). These findings in the more dorsal portion of PMC are notable because there were no significant age-related differences in this region, suggesting that these shifts are not due to overall changes in attention or executive function. This result is in line with our hypothesis that increasing age is associated with more anticipation of higher level themes, and different ages likely process these themes with a different overall pattern of activity.”

Even after removing the subject with high motion, a t-test reveals that there is significantly more motion in the Youngest age group than in the Oldest age group (t(125) = 5.12, p = 1x10-6). We performed standard preprocessing measures to try to minimize the impact of motion. We thus regressed out framewise displacement, the global signals from the cerebrospinal fluid and white matter, eight discrete cosine filters with 128s cut-off, and six rotation and translation parameters, as well as their derivatives as nuisance regressors out of the fMRI time courses. We additionally performed a new analysis to determine whether the result that some parcels have lower ISC in the Youngest group could be explained as an artifact of increased motion. We compared the framewise displacement for each individual in the Youngest group and the correlation of their brain response with the rest of the group (in all parcels where ISC increased with age). We found no relationship between framewise displacement and ISC and now report this in the Results:

“To ensure that the increases in ISC with age are not due to differences in the level of noise between the groups, we measured the relationship between the framewise displacement of each child in the Youngest group and their ISC with the other subjects. There was no relationship between framewise displacement and ISC in any of the parcels where ISC increased with age, indicating that motion did not drive the result in these parcels (all q’s>0.05).”

We therefore conclude that head motion alone cannot explain the difference we observed between age groups.

To further compare the ISC values in the Youngest and Oldest groups, we now include an additional supplementary figure (Supplementary Figure 2) which displays ISC across the cortical surface for both age groups. Both show a similar pattern of ISC that increases from sensory to anterior regions.

To ensure that the between-group ISC measure (Figure 2) is not driven by lower data quality in the younger participants than in the older participants, following the suggestion from Reviewer 2, we have calculated Young-Young ISC minus Young-Old ISC (see Author response image 1). (a) displays this without thresholding for a significant difference between the two quantities and (b) displays this with thresholding for multiple comparisons (q < 0.05).

**Author response image 1. sa2fig1:** 

All significant parcels are positive values, indicating that Young-Young ISC is always greater than Young-Old ISC where it is significant. There is therefore a reliable signal in the young group.We have opted to not include this in the manuscript because ISCb inherently controls for differences in data quality or reliability between the groups by dividing the between-group ISC by the geometric mean of the within-group ISCs. Additionally, the significant regions in panel (b) largely overlap with those found to have significant between group differences in Figure 2.

To better assess any potential differences in data quality that may be driving our HMM fits, we have revised Figure 5—figure supplement 1 (formerly Supplementary Figure 2) to include modelfit differences displayed across the cortex for both the Youngest and Oldest groups.

This illustrates a very similar pattern of HMM fits across cortex for both age groups. Additionally, we have revised the scatterplot portion of the figure such that the x-axis scale equals the y-axis scale, and there is now an x=y line for illustrative purposes. In the methods section, we now report the number of datapoints below the threshold for model-fit difference (0.002) in each group:

“In 40 parcels the Youngest group had a model fit difference less than 0.002 and in 33 parcels the Oldest group had a model fit difference less than 0.002.”

2. It is not clear which hypotheses were tested, supported and refuted in the current study. We suggest you either improve the theoretical framework of the study, and clearly describe the hypotheses tested, or reframe the paper as an exploratory or methods paper. The reviewers provide some suggestions for how you might go about this.

We have now significantly reframed the introduction and discussion portions of the paper in line with improving the theoretical framework of the paper, and in order to better describe the hypotheses tested.

We have now better defined our hypotheses and defined the terms in the manuscript more clearly. In the introduction, we define “schematic event representations” as those that:

“are able to generalize across different instances of similar events”

and we define “episodic encoding processes” as those:

“responsible for encoding the specific details of events.”

We also better outline the regions or networks of regions that we hypothesize to be responsible for these processes. We now write:

“We hypothesized that in default mode network regions responsible for story interpretation and self-referential thought, even where within-age ISC magnitude does not change, the pattern of activity representing the video will change across development, just as the semantic interpretation of videos changes with age (Nelson, 1986; Raichle et al., 2001).”

and

“We hypothesized that with age, the strength of schematic event representations, that are able to generalize across different instances of similar events, will increase due to more experience with different exemplars. These kinds of representations should be stored in default mode regions such as the medial prefrontal cortex and posterior medial cortex (PMC).”

We now also better define the grounds for refuting this hypothesis:

“If we do not find a general improvement in event model fits with age, this will refute the hypothesis that the schematic event representations that support event models are strengthened with age.”

To better describe our hypothesis, we now write:

“In line with the idea that schematic event representations improve with age, older adolescents should be able to anticipate events further into the future due to their increased experience with the world.”

And finally, we clarify what it would mean to refute the hypothesis that the magnitude of the hippocampal response to event boundaries would increase with age, in line with a model of maturation wherein the ability to encode the unique episodes of daily experience increase into middle age and then decreases with senescence,” when we write:

“However, should we find a decrease in the hippocampal response to event boundaries with age, this would provide support for the idea that younger children may focus on the episodic encoding processes responsible for encoding the specific details of events as they work towards creating more stable schematic event representations (Keresztes, et al., 2018; Maril et al., 2010).”

In the discussion, we have clarified where we did not find support for our specific brain-related hypotheses. We now write:

“This finding contrasts with our hypothesis that schematic event representations should increase with age in regions such as the PMC.”

We also have an entire paragraph dedicated to our mixed findings in the medial prefrontal cortex:

“We found small, but significant changes in the timecourse of narrative processing in the medial prefrontal cortex (Figure 2). This region has been previously implicated in tracking schematic knowledge during video-watching (Baldassano et al., 2018; Van Kesteren et al., 2010), suggesting that schematic responses evolve across this age range. However, we surprisingly did not find a change in the strength of event representations with age. This may be due to an overall low level of ISC in this region, either due to idiosyncratic cognition, inconsistent neuroanatomy across subjects, or noisy signals in this region (Supplementary Figure 1), making it challenging to study this region using the group-level approaches in this study. This result is not entirely unexpected given that previous studies have found low levels of ISC in this region (Baldassano et al., 2017), especially in developmental populations (Lerner et al., 2019; Moraczewski et al., 2018). Furthermore, as has been found previously, it may be that only a subregion of the medial prefrontal cortex represents events (Baldassano et al., 2017; Brod et al., 2017; Masís-Obando, Norman, and Baldassano, 2022; Van Kesteren et al., 2010).”

Finally, we clarify what hypothesis the hippocampal (HPC) results support:

“The HPC may thus have a more robust signal in young children who need to focus on encoding the specific details of events as they use them to extract commonalities across experiences (Keresztes, et al., 2018; Maril et al., 2010). Likewise, the HPC has been shown to have increased activity during memory consolidation, and in response to novel events (Nadel, and Moscovitch, 1997; Stern et al., 1996). It is possible that younger children found the events in the video generally more novel, or needed to spend more effort consolidating them due to their impoverished schematic representations. However, if this was truly a novelty-related response we would have expected to see a greater age-related change in the pHPC which has been shown to be more responsive to novelty (Stern et al., 1996).”

3. It would greatly strengthen confidence in the current findings, which seem quite exploratory rather than hypothesis-driven, if they could be replicated in an independent data set. There are multiple public datasets of children watching videos, could any of these be leveraged to replicate the current findings?

We agree that a replication would increase the confidence in the current findings. However, we do not know of a publicly available dataset that meets the criteria necessary for running the analyses in the paper. This dataset is unique in several ways, having both (a) a large number of subjects of each age, and (b) a narrative video stimulus that is long enough to study regions with event timescales that can extend multiple minutes. Fitting a Hidden Markov Model with the minimum number of events (2), requires a video of at least ~six minutes (and preferably ten minutes or more) to access timescales extending up to ~three minutes. We address these limitations in the discussion where we write:

“…our analyses require stimuli that exhibit multiple transitions between events that are at least 30 seconds. We were therefore unable to make use of the other video available in this dataset or data from the short video clips used in other developmental datasets (Alexander et al., 2017; Richardson et al., 2018).”

4. The boundary ratings are obtained from adults, and may therefore not accurately reflect those of younger participants. It would strengthen the paper if boundary ratings could be obtained across the age range studied here.

Obtaining boundary ratings from a developmental population is challenging, since there is currently no standardized protocol for how to conduct this type of experiment. In this revision, we devised an online study to collect boundary ratings from other children whose ages are between 5 and 18 years.

We briefly describe the findings in the Abstract:

“Over the course of development, brain responses became more discretized into stable and coherent events and shifted earlier in time to anticipate upcoming perceived event transitions, measured behaviorally in an age-matched sample.”

And introduce our approach in the Introduction:

“How can we characterize the changes that are occurring in response timecourses? Although some knowledge of the hierarchical structure of the events that compose a narrative develops in infancy, the ability to reliably notice these events does not mature until at least the teenage years (Zacks and Tversky, 2001; Zheng, Zacks, and Markson, L., 2020). It is likely that the characterization of events changes with age. We therefore asked both children and adults to subjectively report where they believed meaningful scene changes occurred in the narrative. We hypothesized that although there would be no change in the behaviorally reported location of these coarse-grained narrative segments, the neural representation of events along the cortical hierarchy would change with age.”

We describe the procedure in the Methods section in the Stimulus and event rating section:

“The perceived timing of event boundaries according to participants within the age range that fMRI was recorded (5-19 years) was assessed online via the Gorilla platform (www.gorilla.sc; Anwyl-Irvine et al., 2020). The online experiment involved a brief training to ensure comprehension of the task. Participants first watched a different clip from Despicable Me (03:25 to 04:47 in the full video). Participants were notified after the first event boundary (04:10 in the full video) and asked to identify the next event boundary (04:37 in the full video). Participants were given three attempts to correctly identify the second boundary. If a participant successfully identified the second boundary, they watched the same clip that was used during fMRI scanning and identified boundaries in that clip at their discretion. The children's data were first analyzed in aggregate, and then analyzed after splitting based on the median age (12.25 years).

The online task was also administered to 25 adults (9 male, 19-56 years) recruited via Prolific (www.prolific.co). The data from two adults were not analyzed as their ages overlapped with the age range of the fMRI participants (they were under 19 years). Data from 66 participants (39 male, 7-18 years) within the age range of the fMRI participants were recruited via word of mouth and Facebook. The data from eight subjects were eliminated as their median event duration was less than one second. All online experimental procedures were approved by the Columbia University IRB (protocol number AAAS0252, for adults, and AAAT8550, for children), and the data is available online: https://github.com/samsydco/HBN.”

And in a new Methods section regarding the analysis of this new behavioral data:

“Analysis of event segmentation behavior

Event segmentation behavioral data from each individual was first convolved with a canonical hemodynamic response function (HRF, one parameter γ model from AFNI's 3dDeconvolve) at each timepoint (Lee et al., 2021). This data was compared between children, adults, and the younger and older children as determined by a median split based on age. Data was compared within and between groups by splitting each group in half, and averaging the data across raters within a split, thus computing shISCs where appropriate. This procedure was done on five random splits of the data. The ISCb measure was used to compare response timecourses between the groups. These values were computed across five random splits of the data and results were averaged across the splits. All significance values were computed on 10,000 random permutations of the data.”

We have added the new data to the Results section, with an accompanying new figure (Figure 3):

“This result demonstrates that responses are changing substantially with age in many brain regions, but does not indicate how these responses are changing. One possibility is that the interpretation of the events and scenes in the narrative changes with age. We therefore assessed when adults and children, age-matched to the fMRI sample, report that an event in the narrative has finished, and a new event has begun. There are small but significant differences in the timing of the event boundaries marked by adults and children (ISCb = 0.97, p = 0.009). This discrepancy was true for both the younger and older children, separated by a median split based on age (adult-older children ISCb = 0.95, adult-younger children ISCb = 0.91, both p’s < 1x10-5). The timing of event boundaries was also slightly, but significantly different between the younger half and older half of the children (ISCb = 0.96, p = 0.01, Figure 3). Given this overall similarity in

behavior, we next ask whether there are differences in how these events are represented and tracked in the brain.”

And we briefly discuss the results in the discussion:

“… behaviorally, we found small, but significant differences in segmentation behavior between adults and children, and children of different ages.”

All analyses that compare brain data with behavior have been updated to compare the brain data with the age-matched behavioral data. We have thus updated Figure 6 and Figure 5—figure supplement 3, and our results and conclusions do not change. We have additionally updated relevant portions of the Methods to reflect that the behavioral data is from an “age-matched” sample.

5. How can we be sure developmental changes in the timing of event transitions reflect increasing "anticipation" in the older group, rather than lagging processes in the younger group? And are anticipatory shifts as long as 20s cognitively plausible?

To better assess whether we are measuring anticipation in the older group or lagging processes in the younger group we have added an additional analysis to the paper. In this analysis we compare the timing of the event transitions in each age group to the behaviorally-derived event boundaries to determine the parcels in which the brain’s activity leads the recognition of behavioral events, and the parcels in which the brain’s activity lags behind the behavioral events. We conducted this analysis separately in the Oldest and Youngest groups and now report on only the parcels with significant event timing differences. We have added these analyses to the Results and Methods sections, as well as displaying them in Figure 6b.

In the Results section, we write:

“To determine whether developmental changes in the timing of event transitions reflect an increased level of anticipation in the Oldest group, or, alternatively, a delayed response in the Youngest group, we compared the HMM-derived event boundaries to event transitions behaviorally identified by an age-matched group of children. Across the cortex we find that the Oldest ages generally anticipate event shifts, while the Youngest ages lag behind them to some degree (Figure 6b).”

And in the Methods section, we write:

“In all parcels that had a significant change in event timing between age groups, we compared the timing of the event boundaries identified by the HMM to the event boundary timecourse obtained behaviorally from raters. Following methods used previously, the number of boundary annotations from the child raters at each timepoint during the video were convolved with a canonical HRF (Lee et al., 2021). To obtain a measure of when the brain is switching between events, we took the derivative of the expected value of the event that the HMM assigned to each timepoint (following methods from Lee et al., 2021). To determine the timing of the alignment between these two signals, we cross-correlated the HMM-derived boundary timecourse with the behavioral event boundary timecourse from children, in aggregate. To precisely estimate the optimal lag, we fit a quadratic function to the maximum correlation lag and its two neighboring lags, and found the location of the peak of this quadratic fit. We computed the maximum-correlation lag separately in the Oldest and Youngest groups to determine whether there is a significant difference in the relationship between the brain’s event boundaries and the behavioral event boundaries.”

This additional analysis thus confirms our interpretation that the Oldest group is in fact anticipating the event transitions in the video ahead of those in the Youngest group. We believe that anticipatory effects as large as 20 seconds are plausible in a 10 minute story in which individual events can last as long as 130 seconds (see Figure 5—figure supplement 2). Previous work has found anticipation up to 15 seconds in a much shorter (90 second) stimulus (Baldassano et al., 2021).

6. Given the relatively uncontrolled nature of the stimuli, it is challenging to know exactly what video content drove responses for any given region. In addition to neural variance, differential eye and head movements induced by move content may also explain variance in the data. This should be acknowledged and discussed.

We agree that there are many factors that may drive the responses to an uncontrolled stimulus such as the one used here. As discussed in our response to Essential Revisions comment #1, we did not find that head motion in the Youngest group had a major impact on ISC values. Eye movements were unfortunately not recorded as part of this dataset, making it unclear the extent to which changes in viewing patterns are driving the observed results in the visual cortex. We have now added this clarification in the Discussion:

“Unfortunately, as eye movements were not recorded, we cannot establish the relationship between eye movements and synchrony in sensory cortex (Alexander et al., 2017).”

7. There is a relatively loose use of anatomy in places, e.g. regarding the TPJ. This should be either improved or acknowledged.

We now define the TPJ more specifically, based on the functional theory of mind ROI from Dufour et al., (2013). We label parcels belonging to TPJ as only those parcels for which two thirds of their area falls within this functional ROI, as described in the Methods section of the paper.

“Defining temporoparietal junction

To better understand how our findings relate to the previous literature we defined the portion of the temporoparietal junction (TPJ) responsible for theory of mind processes based on its definition in Dufour et al., (2013). We projected the definition of the TPJ from this paper, originally in voxel space, onto the cortical surface, and parcels in which over two-thirds of their vertices were within the boundary of the TPJ were labeled as members of this region.”

We have therefore reduced our definition of regions within the TPJ to only three parcels (one in the left hemisphere, and two, relatively smaller parcels in the right hemisphere), illustrated in Figure 2. We have changed the label for the region in Figure 4 to “left parietal operculum”, and we have changed the label for the region in Figure 6 to “left postcentral sulcus.” When we refer to these regions in the discussion, we make it clear which regions overlap with the TPJ, and which do not:

“In the right posterior-superior temporal sulcus (part of TPJ as defined by Dufour and colleagues, 2013) and in the TPJ-adjacent left postcentral sulcus, we found that neural representations in 16-19 year-olds can be as far as 20 seconds ahead of those for 5-8 year-olds on the first viewing of a video clip.”

We encourage you to refer to the individual reviews for detailed recommendations on how to address these points.Reviewer #1 (Recommendations for the authors):– Could increased similarity in participants' eye movements (i.e. the way they overtly sample the videos) with age explain some of the effects reported here?

Yes, an age-related change in the similarity of participants’ eye movements could definitely explain some of the effects that we observe. We highlight this possibility in the discussion where we write:

“Older ages exhibit more across-subject synchrony in visual and auditory cortices… These results are likely related to more strongly correlated semantic processing in adolescents, resulting in top-down increases in synchronous eye movements and auditory understanding.”

Unfortunately, the dataset that we analyzed for this project did not track eye movements during video viewing. We have now added this limitation in the Discussion:

“Unfortunately, as eye movements were not recorded, we cannot establish the relationship between eye movements and synchrony in sensory cortex (Alexander et al., 2017).”

It would have been interesting to see if age-related changes in eye movement synchrony help to explain any of our cortical results.

– Could there be a difference in head motion between age groups? Importantly, head motion might affect some regions (e.g. lateral ones) more than others (medial ones).

Yes! Thank you for bringing this important caveat to our attention. We have now analyzed the difference in framewise displacement between the Youngest and Oldest ages. We report this difference, as well as how we address potential differences in the effect of head motion on lateral regions in which ISC increases with age in our response to Essential Revisions comment #1.

– Do the authors think that any of their effects are due to general development of the cortex rather than narrative comprehension? That is, might some of their effects also be found when participants watch scrambled videos without structure or meaning?

This dataset does not provide a way to separate the effects of the maturation of the brain from the increased experience with the world that underlies schematic knowledge. We now highlight this in the discussion, where we write:

“Furthermore, we cannot distinguish whether the improvements that we see with age are due to the overall maturation of the brain or the increased life experience that underlies schematic knowledge.”

It is unclear how we would interpret data from a scrambled-video paradigm in children, since there would likely be large attentional differences across ages and it would be unlikely to see ISC in any region outside of sensory cortex (see: Hasson, et al., 2008, A hierarchy of temporal receptive windows in human cortex for evidence in adults). Videos without structure and meaning, such as Inscapes (Vanderwal et al., 2015), have been previously used to measure differences in functional connectivity, but are unlikely to evoke the kinds of stable event patterns necessary for applying our Hidden Markov Model analyses.

– Figure 6, right panels: can you make the different age groups more easily distinguishable?

We have increased the thickness of the lines in Figure 6 (now Figure 7 in paper), right panels, and hope that this solves the problem.

– Seimen's -> Siemens

Thank you for alerting us to this typo. We have corrected this in the manuscript.

Reviewer #2 (Recommendations for the authors):Major 1. The authors should significantly reframe the paper. There are many ways this could be done. One possibility is to hone attention toward a specific network most closely associated with "narrative" processing based on similar adult work, or which shows the strongest correlations to behaviorally-determined event boundaries. In this case, the authors should take care to provide clear definitions of the various terms in the manuscript, and clear hypotheses and alternative hypotheses. A second possibility is that the paper could be reframed in terms of the methodological approach (i.e., using HMM to explore event structure in naturalistic video data, which may have particular promise in pediatric datasets).

We have taken the reviewer’s first suggestion to better define our hypotheses and define the terms in the manuscript more clearly. We outline these specific changes in our response to Essential Revisions comment #2.

Major 2. Specific suggestions for addressing the concern that key effects are driving by lower data quality in younger children:The authors should show within-group ISC across the cortex for young and old groups separately, not just the direct comparison of the two groups. Beyond confirming that the data in young children look reasonable, this analysis would reveal which regions actually show reliable within-group ISC (especially in the young group alone), and which do not. For example, it is possible that only occipital parcels show strong within-group ISC (as suggested by Supplemental Figure 1); if so, then it is not surprising to find no difference in within-group ISC between older and younger groups in regions beyond the occipital cortex, since reliable video responses could not really be detected there.For the new between-group ISC measure (Figure 2), despite normalizing by within group ISC, this result could still be driven by lower data quality in younger kids than older kids. The strongest test of a qualitative developmental change in video responses would be to ask whether "young-young group" ISC is greater than "young-old group" ISC. If young children predict other young children better than older adolescents, then there is a reliable signal in the young group that cannot be explained by lower data quality in the younger sample than the older one. The alternative is that any comparison between a good dataset and a bad dataset will yield lower between group ISC than expected based on within-group ISC (i.e., driven by the older group only, by virtue of the higher data quality in that group).The authors should also show HMM fits across the cortex, particularly in the young group alone. The comparison of model-fit-difference between young and old subjects in Supplemental Figure 2 is encouraging, although it appears that the young and old data are presented on slightly different scales (x versus y axis), and that more parcels did not show significant model fits in young children than older adolescents (based on the number of data points below the box "cutoff" line on each axis; the authors should report this number for each group directly, rather than the combined measure across groups). Further, in Figure 4 (side panels), it seems that the optimal timescale for the youngest group in many regions (TPJ, V1, A1) barely beats out the longest timescale, again calling into question the quality of the young child data. Finally, it would be helpful to know which parcels are showing the best model fit, and which show only weak model fits, again in both young and old groups separately.

We address these points with several additional figures and analyses in our response to Essential Revisions comment #1.

Reviewer #3 (Recommendations for the authors):Per my public review, I do not have what I would consider to be serious issues with the manuscript. In general, I think this is very good science and will make a solid contribution. I do, however, have some concerns that would make for a stronger publication if addressed:(1) The authors obtained independent behavioral boundary ratings from adults, and compared these ratings to the neural responses from the participants. It does not appear that any of the independent raters were in the actual age range of interest (5-19 years). I suggest obtaining independent behavioral boundary ratings from the same age group to determine whether neural responses track with the behavior of the same age group. I will note that this discrepancy is not, in my view, a huge issue for the paper either way. Rather, I think there are interesting and important questions pertaining to this issue, namely whether behavioral measures pertaining to event boundaries differ fundamentally across this age range, and whether this has any bearing on the neural measures obtained in this study. It would be an equally novel and important contribution to demonstrate that this either does or does not have major influences on the patterns of results one might observe. While I am hesitant to request further data collection and/or analyses, in this case, I think it is warranted and would strengthen the kinds of conclusions that can be drawn from these data.

We have now collected additional boundary ratings from children and we discuss these new results in our response to Essential Revisions comment #4.

(2) On page 11, the authors suggest that the decreased response reliability, anticipation, and model fit performance is possibly due to the PMC performing "additional processing in the absence of strong schemas… [and] younger children may therefore more consistently rely on this region to understand a relatively more novel environment." It is unclear to me why people would anticipate an unfamiliar scene transition. Furthermore, based on the authors' prior work, it is unclear why the PMC, if it is relatively sensitive to higher-order event schemas (Baldassano, Hasson, and Norman, 2018), should show more anticipation by younger children. I am afraid I do not understand the logic here, or how to reconcile it with other findings. Can the authors clarify this?

After eliminating one subject whose motion was three standard deviations above the rest of the subjects, the PMC no longer significantly decreases in anticipation. The RSC still decreases in reliability with age (Figure 1), and the PMC does decrease in model fit with age (Figure 5). An updated version of Figure 5 is now Figure 6. It seems that now, if anything, the left PMC actually increases in anticipation with age, as we would expect.

We interpret this result in the discussion, where we write:

“However, we did find both age-related shifts in the pattern by which the narrative was represented bilaterally (Figure 2) and increased anticipation in the left PMC (Figure 6). These findings in the more dorsal portion of PMC are notable because there were no significant age-related differences in this region, suggesting that these shifts are not due to overall changes in attention or executive function. This result is in line with our hypothesis that increasing age is associated with more anticipation of higher level themes, and different ages likely process these themes with a different overall pattern of activity.”

We have also added additional interpretation to the finding that event model-fits in PMC decrease with age in the discussion:

PMC has an intrinsically long timescale for information processing, one that emerges in infancy, which may thus support strong event models in children as young as five (Hasson et al., 2015; Stephens, Honey, and Hasson, 2013; Yates et al., 2021).

(3) A general concern when comparing neural signals across different age ranges is the issue of signal quality, which can differ markedly across groups. The authors have seemingly accounted for potential differences in SNR in several formulae in the appendices, but do not mention in the main text whether age-related differences in SNR were examined or accounted for. This may perhaps be deducible from the formulae the authors provide, but regardless, I think a more direct mention of this issue would be helpful.

We address potential differences in SNR between the groups in our response to Essential Revisions comment #1.